# Mixed-mode oscillations and population bursting in the pre-Bötzinger complex

**Bartholomew J Bacak[1]\*[†], Taegyo Kim[1][†], Jeffrey C Smith[2], Jonathan E Rubin[3], Ilya A Rybak[1]**

[1]Department of Neurobiology and Anatomy, Drexel University College of Medicine, Philadelphia, United States; [2]Cellular and Systems Neurobiology Section, National Institute of Neurological Disorders and Stroke, National Institutes of Health, Bethesda, United States; [3]Department of Mathematics, University of Pittsburgh, Pittsburgh, United States

**Abstract** This study focuses on computational and theoretical investigations of neuronal activity arising in the pre-Bötzinger complex (pre-BötC), a medullary region generating the inspiratory phase of breathing in mammals. A progressive increase of neuronal excitability in medullary slices containing the pre-BötC produces mixed-mode oscillations (MMOs) characterized by large amplitude population bursts alternating with a series of small amplitude bursts. Using two different computational models, we demonstrate that MMOs emerge within a heterogeneous excitatory neural network because of progressive neuronal recruitment and synchronization. The MMO pattern depends on the distributed neuronal excitability, the density and weights of network interconnections, and the cellular properties underlying endogenous bursting. Critically, the latter should provide a reduction of spiking frequency within neuronal bursts with increasing burst frequency and a dependence of the after-burst recovery period on burst amplitude. Our study highlights a novel mechanism by which heterogeneity naturally leads to complex dynamics in rhythmic neuronal populations.

**\*For correspondence:**
BartBacak@gmail.com

[†]These authors contributed equally to this work

**Competing interests:** The authors declare that no competing interests exist.

## Introduction

Mixed-mode oscillations (MMOs) represent rhythmic behaviors of dynamical systems characterized by an alternation between large amplitude (LA) and small amplitude (SA) oscillations (*Desroches et al., 2012*) and have been observed in many physical, chemical, and biological systems, including a variety of neural structures. The latter include populations of neurons in the entorhinal cortex (*Dickson et al., 2000*; *Yoshida and Alonso, 2007*), hippocampal neurons (*Winson, 1978*), dopaminergic neurons (*Medvedev et al., 2003*; *Medvedev and Cisternas, 2004*), neurons of the medullary pre-Bötzinger complex (*Del Negro et al., 2002c*) vibrissa motoneurons (*Golomb, 2014*) and spinal motoneurons (*Iglesias et al., 2011*) in rats.

Theoretical investigations of MMOs typically focus on the mechanisms by which MMOs emerge from a complex interplay of multiple distinct time scales in the nonlinear processes governing a system's activity (*Desroches et al., 2012*). In this work, we introduce and explain a novel alternative paradigm for the generation of MMOs. The key element in the mechanism that we present is that a network of coupled oscillators can generative repetitive MMOs based on heterogeneity within the network. The importance of this paradigm for neural systems relates to central pattern generators (CPGs) that can intrinsically generate rhythmic activity controlling different motor behaviors such as breathing and locomotion. Heterogeneity in the quantitative features of the neurons involved is likely a ubiquitous property of such circuits (*Butera et al., 1999b*; *Marder, 2011*; *Buzsáki and Mizuseki, 2014*), and thus our work predicts that MMO patterns should be attainable in a wide range of

**eLife digest** Each breath we take removes carbon dioxide from the body and exchanges it for oxygen. A structure called the brainstem, which connects the brain with the spinal cord, generates the breathing rhythm and controls its rate. While this process normally occurs automatically, we can also control our breathing voluntarily, such as when singing or speaking.

Within the brainstem, a group of neurons in the area known as the pre-Bötzinger complex is responsible for ensuring that an animal breathes in at regular intervals. Recordings of the electrical activity from slices of brainstem show that pre-Bötzinger neurons display rhythmic activity with characteristic patterns called "mixed-mode oscillations". These rhythms consist of bursts of strong activity ("large amplitude bursts"), essential for triggering regular breathing, separated by a series of bursts of weak activity ("small amplitude bursts"). However, it is not clear how mixed-mode oscillations arise.

Bacak, Kim et al. now provide insights into this process by developing two computational models of the pre-Bötzinger complex. The first model consists of a diverse population of 100 neurons joined by a relatively small number of weak connections to form a network. The second model is a simplified version of the first, consisting of just three neurons. By manipulating the properties of the simulated networks, and analysing the data mathematically, Bacak, Kim et al. identify the properties of the neurons that allow them to generate mixed-mode oscillations and thus rhythmic breathing.

The models suggest that mixed-mode oscillations result from the synchronization of many neurons with different levels of activity (excitability). Neurons with low excitability have low bursting frequencies, but generate strong activity and recruit other neurons, ultimately producing large amplitude bursts that cause breathing.

Many parts of the nervous system are also made up of networks of neurons with diverse excitability. A challenge for future studies is thus to investigate whether other networks of neurons similar to the pre-Bötzinger complex generate rhythms that control other repetitive actions, such as walking and chewing.

brain structures with rhythmic activity depending on mechanisms for neuronal synchronization. Furthermore, predictions that follow from the existence of this MMO-generation mechanism should be of similarly widespread relevance.

For concreteness, the present study focuses on computational models of a neuron population in a particular brain area, the pre-Bötzinger complex (pre-BötC), where MMOs have been previously observed (*Del Negro et al., 2002c*). The pre-BötC is a medullary region representing an excitatory kernel circuit of the respiratory CPG in mammals that is critically involved in generating the inspiratory phase of respiration (*Smith et al., 1991*; *Smith et al., 2007*; *Smith et al., 2009*; *Smith et al., 2013*). The pre-BötC can generate rhythmic bursting activity *in vitro*, in medullary slices containing this structure (*Koshiya and Smith, 1999*; *Del Negro et al., 2001*; *Rigatto et al., 2001*; *Thoby-Brisson and Ramirez, 2001*) and even in *isolated* 'islands' extracted from these slices (*Johnson et al., 2001*; *Figure 1A*). This rhythmic activity is typically induced by elevating the extracellular concentration of potassium ($[K^+]_{out}$) up to 7–9 mM, which putatively increases neuronal excitability (*Koshiya and Smith, 1999*; *Lieske et al., 2000*; *Del Negro et al., 2001*; *Johnson et al., 2001*; *Thoby-Brisson and Ramirez, 2001*). Pre-BötC neurons, through a pre-motor population, project to the hypoglossal nuclei containing motor neurons, the activity of which can be recorded in rhythmically active slices from the hypoglossal (XII) nerve (see *Figure 1*, panels A, B, and C1). Simultaneous optical recordings from individual neurons and XII output have shown that bursts in the XII root represent the synchronized activity of pre-BötC neurons (*Koshiya and Smith, 1999*; *Figure 1C1,C2*) and the amplitude of XII bursts clearly depends on the number of pre-BötC neurons involved. Interestingly, a progressive increase in $[K^+]_{out}$ in slices containing the pre-BötC evokes complex population MMOs characterized by amplitude modulation, with large amplitude (LA) bursts alternating with a series of small amplitude (SA) bursts (*Koshiya and Smith, 1999*; *Del Negro et al., 2002c*; *Kam et al., 2013*) (see *Figure 1A*, bottom). An amplitude irregularity similar to the MMOs recorded from the pre-BötC *in vitro* has also been observed during acute intermittent hypoxia

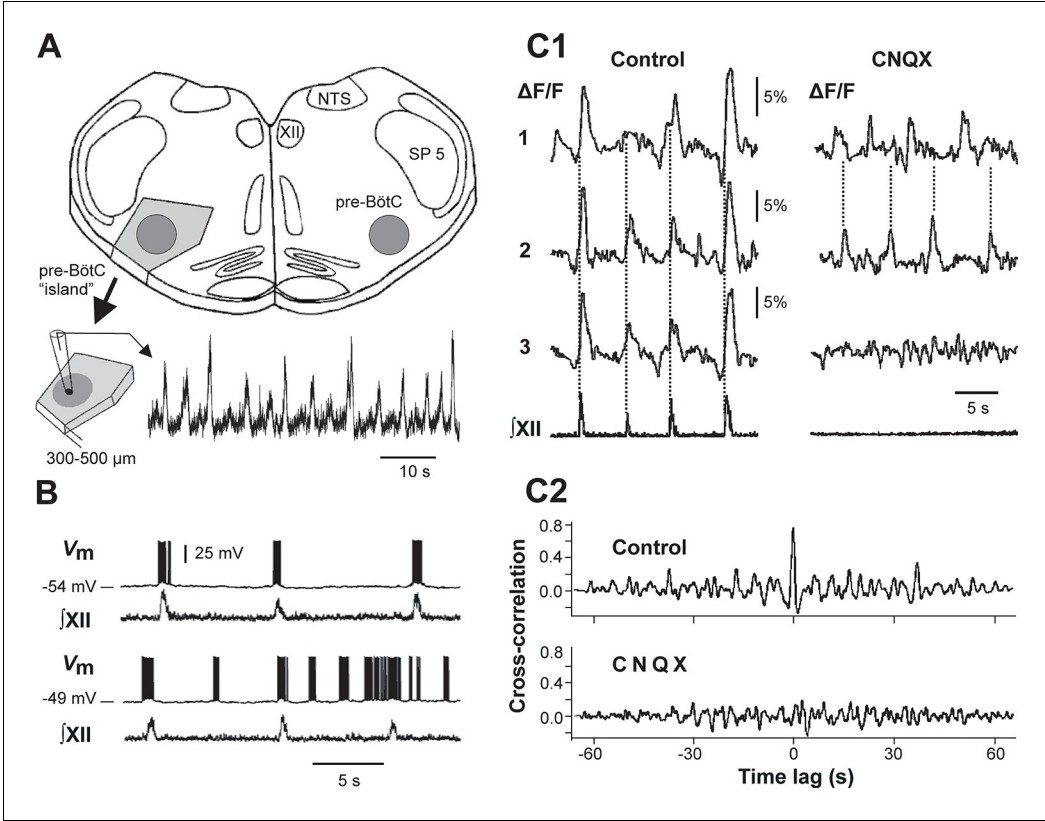

**Figure 1.** Mixed-mode and endogenous oscillations in the pre-Bötzinger complex *in vitro*. (A) Top: medullary slice showing 'pre-BötC island' (shaded dark gray) and labeled structures: XII, hypoglossal motor nucleus; NTS, nucleus tractus solitarius; SP 5, spinal trigeminal tract. Bottom: Excised pre-BötC island with extracellular recording from the pre-BötC that demonstrates MMOs (i.e., interleaved large and small amplitude bursts). Modified from *Johnson et al. (2001)*. (B) Intracellular recording from pre-BötC neuron with baseline membrane potential of -54 mV (top trace) and -49 mV (bottom trace). The corresponding integrated hypoglossal motor output (∫XII) is shown below each neuronal recording. In the top trace, each neuronal burst coincided with the activity in the hypoglossal motor output. At the more depolarized baseline potential, bursting occurred at higher frequency and several ectopic bursts did not correspond to ∫XII output. (C1) Optical recording from pre-BötC neuron activity ($Ca^{2+}$ imaging). Left: three inspiratory neurons (1–3) show synchronized $Ca^{2+}$ activities (ΔF/F) and corresponding ∫XII output (synchronization marked with dotted lines). Right: Application of CNQX (6-cyano-7-nitroquinoxaline-2,3-dione, blocking fast glutamatergic synaptic transmission, 50 μM) caused a loss of bursting in ∫XII and neurons 1 and 2 showed desynchronized bursting activity (see dotted lines). (C2) Cross-correlograms for neurons 1 and 2 in C1. The loss of a peak at 0 time lag after CNQX indicates loss of synchronization. B, C1, and C2 were adapted from *Koshiya and Smith (1999)*.

simulated *in vitro* (*Zanella et al., 2014*). Similar pathological patterns of breathing have been observed in vivo in association with different diseases, such as myocardial infarcts, obstructive sleep apneas, apneas of prematurity, Rett syndrome, and sudden infant death syndrome (*Zanella et al., 2014*).

To theoretically investigate the mechanisms underlying these MMOs, we developed and analyzed two models: (a) a computational model of a network of 100 neurons, described in the Hodgkin-Huxley style, with bursting properties defined by the persistent (slowly inactivating) sodium current ($I_{NaP}$) incorporated in each neuron, with sparse excitatory synaptic interconnections, and with randomly distributed neuronal parameters, and (b) a simplified model consisting of three mutually excitatory non-spiking neurons that allowed us to apply qualitative analytical methods for understanding key system behaviors. Our simulations and analysis suggest that neurons with low excitability, which generate low frequency bursting with high intra-burst spike frequency, recruit LA bursts by synchronizing the activity of many neurons in the network and therefore play a critical role in the generation of

MMOs. Our simulations and analysis of these models provide important insights into how heterogeneity of neural excitability and other network features contribute to the generation of rhythmic activities in neuron populations that are key components of central pattern generators in vertebrates.

## Results

### Computational modeling of a network of pre-BötC neurons with sparse excitatory synaptic interconnections

Intracellular recordings from individual pre-BötC neurons in rhythmically active slices show a range of resting membrane potentials and other quantitative properties among individual neurons (*Del Negro et al., 2001*; *Del Negro et al., 2002a*; *Peña et al., 2004*; *Koizumi and Smith, 2008*). Neurons with more negative resting membrane potentials usually generate bursting activity that is fully consistent with, and reflected in, XII output activity, whereas neurons with less negative resting membrane potentials demonstrate higher burst frequencies and often generate 'ectopic' busts not reflected in the XII output (see example in *Figure 1B*). Pharmacological blockade of synaptic transmission within the pre-BötC by 6-cyano-7-nitroquinoxaline-2,3-dione (CNQX) results in a reduction and desynchronization of neuronal activity within the pre-BötC, with no activity in the hypoglossal output (see example in *Figure 1C1,C2*).

In light of these experimental findings, we modeled the pre-BötC as an excitatory network consisting of 100 neurons described in the Hodgkin-Huxley style, with sparse excitatory synaptic interconnections between neurons. The intrinsic bursting properties of these neurons were based on the persistent (slowly inactivating) sodium current, $I_{NaP}$ (*Butera et al., 1999a*; *Butera et al., 1999b*; *Del Negro et al., 2001*; *Rybak et al., 2003a*; *Rybak et al., 2003b*; *Rybak et al., 2004*; *Rybak et al., 2014*; *Dunmyre and Rubin, 2010*; *Jasinski et al., 2013*; see *Materials and methods*). To account for neuronal heterogeneity, we distributed the reversal potential of the leak current, $E_L$, across the population (see *Materials and methods* and *Table 1*). We also included mild variability in the maximal conductance of $I_{NaP}$ ($\overline{g}_{NaP}$, *Table 1*), with a range of values that allowed all neurons to be conditional bursters. In the absence of coupling (when all weights of connections were set to zero), the population contained silent neurons, as well as neurons with bursting and tonic activities (*Figure 2A1*). *Figure 2A2* presents the raster plot of neuronal activity in the same population, in which neurons were sorted in order of increasing (from bottom to top) excitability (defined by the assigned $E_L$). This figure shows that neurons with the most negative $E_L$ values were silent (neurons with ID numbers from 1 to 49), neurons with intermediate $E_L$ exhibited bursting activity with burst frequency increasing with $E_L$ (neurons 50–94), and neurons with greatest $E_L$ displayed tonic spiking (neurons 95–100). The lack of network interactions resulted in asynchronous neuronal activity and the corresponding integrated population histogram lacked phasic modulation (*Figure 2A3*).

The patterns of population activity and integrated output dramatically changed when relatively weak and sparse excitatory synaptic connections among neurons were incorporated in the model (*Figure 2B1–B3*). The raster plot of the same sorted neurons in this coupled case (*Figure 2B2*) shows the presence of overlapping clusters (sub-populations) of neurons with synchronized bursting, which generate MMOs characterized by alternating LA and SA population bursts (*Figure 2B3*).

*Figure 3* shows another example of our simulations, including 'uncoupled' (*panel A1*) and coupled (*panel B1*) cases for sorted neurons of the same populations and the integrated population activity for the coupled case (*panel C1*). In both *A2* and *B2* panels we plotted the membrane potentials (V) of four selected representative neurons that in the uncoupled case exhibited (bottom-up): silence (trace 1), bursting with low burst frequency (trace 2), bursting with higher burst frequency (trace 3), and tonic spiking (trace 4). Also in these figures, the time course of the $I_{NaP}$ inactivation variable ($h_{NaP}$) of each neuron, which defined the burst recovery period, was superimposed onto its V time course (red trace). An important feature of all neurons operating in bursting mode is illustrated in *Figure 3A3* (uncoupled case): while the burst frequency (blue curve) increased with the neuronal excitability (bottom-up), the spike frequency within the burst (red curve) changed in an inverse manner, i.e., decreased with increasing neuronal excitability. This reduction of spike frequency within the bursts in more excited neurons limited their ability to synchronize and recruit other neurons' activity in the coupled case (see below).

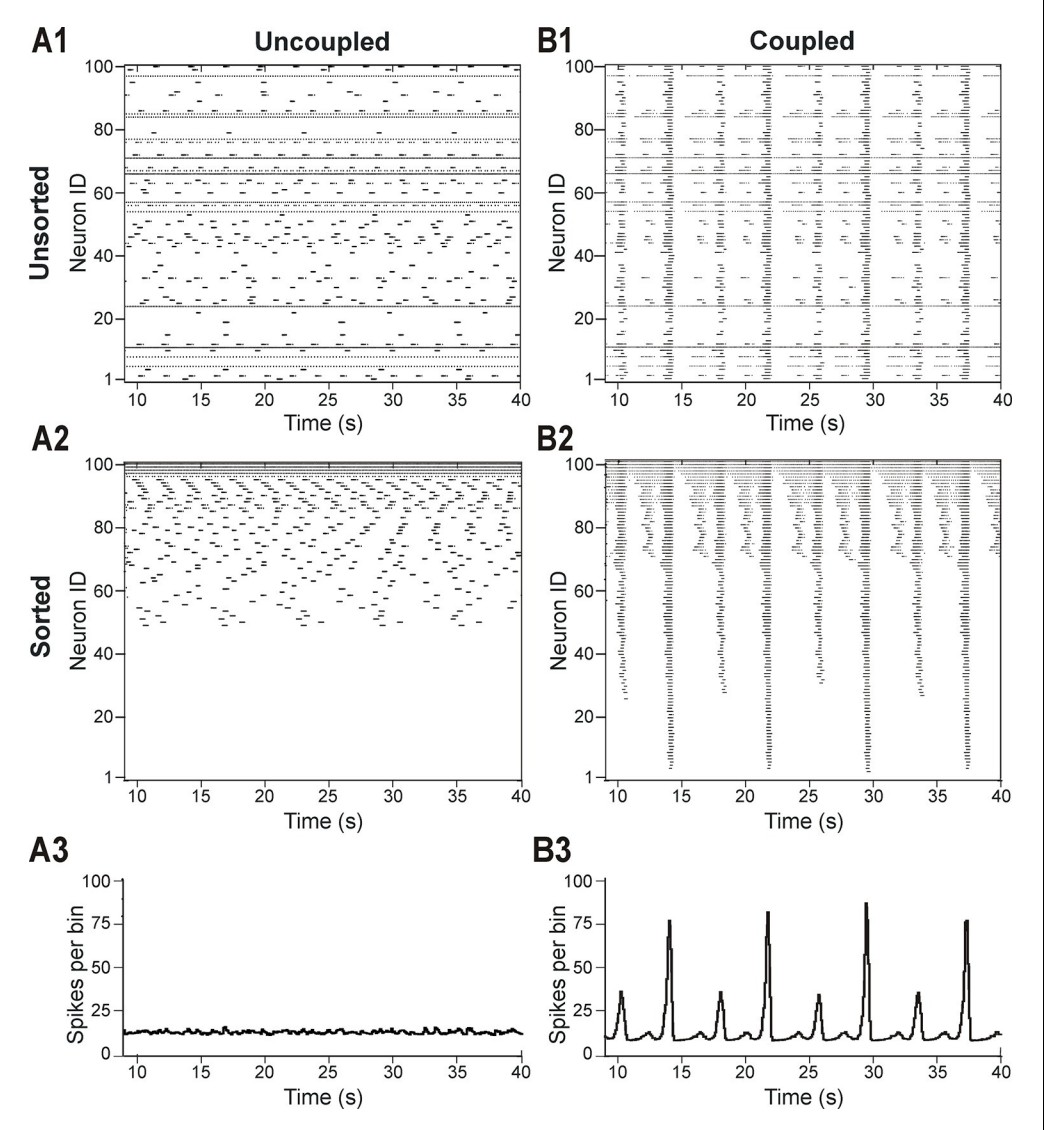

**Figure 2.** Distribution of neural excitability in a sparsely connected network causes mixed-mode oscillations. (**A1-A3**) Network simulation when excitatory interactions between neurons were removed (*w*=0, 'uncoupled' network). (**B1-B3**) Simulation of network activity with sparse excitatory synaptic interconnections at *w*=2.5 and *p*=0.15. (**A1, B1**) Unsorted raster plots depicting the timing of action potentials in neurons with randomly distributed $E_L$ values. (**A2, B2**) Raster plots with neurons sorted by $E_L$ values such that the lowest Neuron ID was assigned to the neuron with the most negative $E_L$. (**A3, B3**) Histogram of population activity. No phasic component is observed in **A3** due to the desynchronized bursting in the uncoupled population. **B3** shows a typical MMO pattern including LA bursts alternating with SA bursts of varying amplitudes.

The pattern of population activity in a coupled network is shown in *Figure 3B1–B3,C1*. Several clusters of neurons with synchronous bursting activity emerged dynamically in the population. Clusters differed by the number of the population bursts in which they participated (*panel B1*), which in turn defined the amplitude of integrated population bursts (*panel C1*). The same panels also show that several relatively small, distinct or partly overlapping clusters with synchronous bursts were formed by neurons with relatively high (less negative) $E_L$. These clusters generated a series of high-frequency SA bursts. Generation of low-frequency LA bursts involved synchronization of many neurons and included those with low excitability (most negative $E_L$) (*Figure 3B1,B2,C1*).

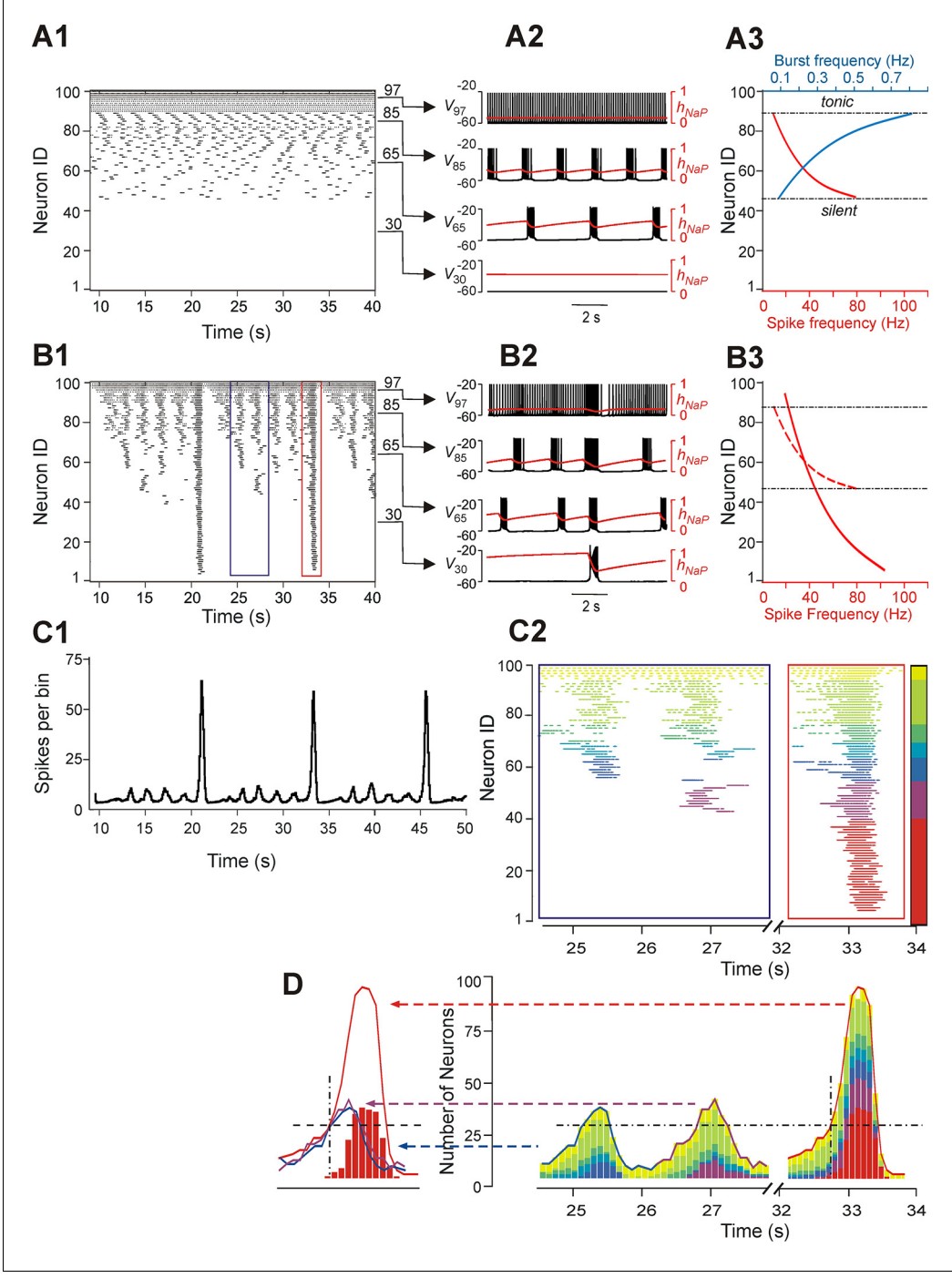

**Figure 3.** Neurons with similar excitabilities activate in clusters within a heterogeneous network with sparse connectivity. (**A1–A3**) Simulation results for uncoupled network, $w=0$. (**A1**) Sorted raster plot showing silent (most negative $E_L$, lowest Neuron IDs), bursting, and tonic (least negative $E_L$, highest Neuron IDs) neurons. (**A2**) Endogenous activity of four individual neurons (Neuron IDs: 30, 65, 85 and 97) showing membrane potential (black) and inactivation, $h_{NaP}$, of the persistent sodium current (red). (**A3**) Burst frequencies (blue) and intra-burst spike frequencies (red) were calculated for each neuron in the uncoupled case. Boundaries separating bursting from silent and tonic neurons are marked (black, dot-dashed lines). (**B1–B3**) Simulation results for $w=1.8$ and $p=0.15$. (**B1**) Raster plot sorted by $E_L$ (Neuron ID). Two SA bursts (blue rectangle) and one LA burst (red rectangle) were selected for the inset in **C2**. (**B2**) Membrane potential (black) and $h_{NaP}$ (red) are shown for the four neurons originally selected in **A2**. (**B3**) Spike frequency of neurons sorted by excitability in the coupled ($w=1.8$) case (the dashed red curve shows spike frequencies for the uncoupled ($w=0$) case in **A3**, for comparison). (**C1**) Histogram of

*Figure 3 continued on next page*

*Figure 3 continued*

population activity corresponding to **B1**. (**C2**) Insets depicting magnified raster plots from the selected bursts in **B1**. Different color families were used to identify neurons that belong to different clusters, with each cluster defined as a group of neurons that participated in the same set of bursts. The clusters of neurons with lowest excitability (*LE*), contributing to only LA bursts, were highlighted by red; on the other side of the 'color spectrum', the neurons with the highest excitability, exhibiting sustained activity, were colored yellow. (**D**, right) The color-coding scheme from **C2** was used in conjunction with a histogram depicting the number of active neurons within a 100 ms window. The vertical dot-dashed black line marks the time of onset of *LE* neuron activation in an LA burst and the horizontal dot-dashed line intersects this onset time to show the total number of neurons already active at the time of *LE* activation. The horizontal dot-dashed line is extended to the two SA bursts and demonstrates that *LE* activation failed despite the presence of a sufficient number of active neurons in the network. (**D**, left) Comparison of number of neurons active over time from two SA bursts (purple and blue curves) and one LA burst (red curve). The intersection of the two dashed, black lines compares the SA and LA burst amplitudes when the *LE* neurons (red bars) first start to activate in an LA burst.

*Figure 3C2* shows two insets from the raster plot in *Figure 3B1* that correspond to two SA bursts (left) and one LA burst (right). The neuronal clusters in these insets are colored as follows: spikes of neurons with default tonic spiking - yellow; spikes of neurons involved in SA bursts - light and dark green, light and dark blue, and purple, arranged in order of increasing excitability; spikes of neurons involved only in LA bursts - red.

The left inset (within the blue rectangle) in *Figures 3C2 and 3D* depicts spikes in the raster plot corresponding to two SA bursts. Two clusters of high excitability neurons, colored by yellow and green, participated in both of these bursts. In addition, the blue cluster participated in the first, but not the second, SA burst, and a purple cluster participated in the second, but not the first, SA burst. The neurons belonging to the red cluster were only active during LA bursts (see right inset within the red rectangle).

To evaluate the role of different clusters in SA and LA bursts in both insets, we built integrated histograms showing the number of neurons, from each colored cluster, that were active within a 100 ms bin (*Figure 3D*). Note that the sub-population of low excitability neurons, colored red, do not contribute to SA bursts. Activation of this sub-population during the LA burst is marked by a black, dot-dashed, vertical line at about 32.7 s. This vertical line intersects with a black, dashed, horizontal line indicating a threshold for the activation of red neurons. This line intersects the two SA bursts demonstrating that, although the amplitudes of both SA bursts rose above the marked threshold for activation of the red sub-population in the LA burst, the latter neurons were not recruited in SA bursts (note the absence of the red neuron cluster in SA bursts) and hence the full LA burst did not develop. We further find that the sub-population of neurons with low excitability (colored red) cannot be recruited by other sub-populations (participating in SA bursts), and hence cannot generate LA bursts, until sufficient recovery of bursting capability in the low excitability neurons (defined by the $I_{NaP}$ inactivation variable $h_{NaP}$) has occurred. This observation suggests that with fixed parameter values, even though the low excitability neurons do not burst when uncoupled, the generation of LA bursts and the durations of their interburst intervals (IBIs) are mostly defined by the operation of an intrinsic burst-supporting mechanism in the less excitable neurons, rather than by variations in the intensity of their recruitment by the activity of highly excitable neurons involved in SA bursts.

## Parameter dependence of mixed mode oscillations (MMOs)

To study the dependence of MMOs on neuronal interactions within the network, we observed changes in the network activity when the weights and/or probability of synaptic connections were varied across simulations. *Figure 4A1,A2,A3* shows three heat maps that demonstrate quantal changes in MMO regimes defined by ratios of LA to SA bursts (e.g. 1:5, 1:4, etc.) as several key parameters were varied (*Figure 4A1,A2,A3*). When either weights (*Figure 4A1,A2,B1*) or probability of connections (*Figure 4A1,A3,B2*) were increased, the frequency of LA bursts increased and the number of SA bursts between successive LA bursts decreased. This corresponded to a progressive change in the quantal state of the network toward regimes with high LA to SA burst ratios.

*Figure 4B1* shows regimes observed when the probability of connections was fixed (*p*=0.15) and only the weights of connections were varied. At the lowest weights (*w* = 1.0), only irregular SA bursts

were observed because of insufficient neuronal synchronization (top trace, *Figure 4B1*). Weights between 1.0 and 1.8 caused regimes characterized by low-frequency irregular LA bursts with irregular patterns of SA bursts (not shown). At a weight of 1.8, each LA burst emerged regularly following five SA bursts (second trace); no parameter sets produced stable regimes with more than five SA bursts per one LA burst. Further increases in weights caused a quantal increase of LA frequency and the corresponding reduction in the number of SA bursts between LA bursts (traces 2–4), until strong enough weights yielded LA bursts only (trace 5). A similar trend is seen in *Figure 4B2* with increases in the probability of connections at a fixed value of synaptic weights ($w$ = 1.8). Overall, for fixed connection weights, the availability of $I_{NaP}$ in low excitability neurons still selects the cycles on which LA bursts occur during MMOs. Furthermore, our simulations showed increased IBIs following the LA bursts, relative to IBIs observed after SA bursts, in all instances of MMOs (*Figures 2B3,3C1,* and *4B1,B2*). In the next section, **Reduced model analysis of interburst intervals** (**IBIs**), we use a reduced model to explain these effects.

Finally, to study the dependence of MMOs on $I_{NaP}$, we varied the average maximal conductance for $I_{NaP}$ ($\overline{g}_{NaP}$) and either weights (*Figure 4A2*) or probability of connections (*Figure 4A3*). The resulting heat maps show a qualitatively similar pattern where the ratio of LA to SA bursts decreases as $\overline{g}_{NaP}$ is reduced. Activity traces corresponding to $\overline{g}_{NaP}$ changes at fixed weights and probability of connections are shown in *Figure 4B3* ($w$ = 3.0, $p$=0.24). At the typical value of $\overline{g}_{NaP}$ (5.0 nS), network activity consisted entirely of LA bursts (*Figure 4B3*, top trace). When $\overline{g}_{NaP}$ was reduced, a decrease in LA burst frequency and an increase in SA burst count between LA bursts were observed (traces 2–4) until busting fully stopped at $\overline{g}_{NaP}$= 3.2 nS (trace 5). Thus, while raising the weights or probability of synaptic connections can enhance the rate of LA burst generation in some parameter regimes, if there is insufficient availability of burst-supporting current, then the recruitment of low excitability neurons is precluded.

## MMOs in a reduced model

A reduced model was developed to allow qualitative mathematical analysis of the MMOs that we observed. The model consisted of three neurons with mutual excitatory synaptic interactions (see *Figure 5A1*). It was considered that each model neuron represented a sub-population of spiking neurons with a particular level of excitability. Each neuron was described using a non-spiking, activity-based model (*Rubin et al., 2009b*; *Rubin et al., 2011*; *Molkov et al., 2015*; see *Materials and methods*). The behavior of each neuron was defined by two dynamical variables, the membrane voltage, $V$, and $I_{NaP}$ inactivation, $h_{NaP}$. For each neuron we calculated a nonlinear output function, $f(V)$, which approximated the aggregate activity of a cluster of neurons in the original 100-neuron model. $E_L$ values were distributed such that in the absence of coupling, neuron 1 (high excitability, *HE*) engaged in high frequency bursting, neuron 2 (moderate excitability, *ME*) engaged in low frequency bursting with no special frequency relation to the bursting of the *HE* neuron, and neuron 3 (low excitability, *LE*) was silent; the three neurons' summed activity provided a representation of network output (*Figure 5B1,C1,D1,E1*). For each simulation, in addition to voltage and summed activity time courses, we visualized the network trajectory as it evolved in ($h_{NaP1}$, $V_1$, $V_3$)-space. Without coupling, this trajectory was cyclic, corresponding to the oscillations of the *HE* neuron 1 (i.e., of $h_{NaP1}$, $V_1$) without changes in $V_3$ (*Figure 5B2*).

In subsequent simulations, neurons in this model interacted through excitatory synaptic interconnections with the weights of connections increasing top-down in *Figure 5* from *panels B1-B3* to *panels E1-E3*. Similarly to the previous model, when connection weights were progressively increased, the network underwent a series of regime transitions progressing from only SA bursts (*Figure 5B1, B2*) to only LA bursts (*Figure 5E1,E2*). The intermediate regimes (*Figures 5C1,C2 and 5D1,D2*) are referred to as 'quantal' and labeled as 1:N regimes if there were N-1 SA bursts between each pair of LA bursts; these correspond to the MMOs in the 100-neuron model described above. The periods of oscillations were calculated for all neurons as weights of connections were gradually increased (*Figure 6A*), and these clearly distinguished the different quantal states observed. As in the previous model, LA bursts involved activation of all neurons and occurred exactly on the cycles when the *LE* neuron activated (*Figure 5C1,C2,D1,D2,E1,E2*).

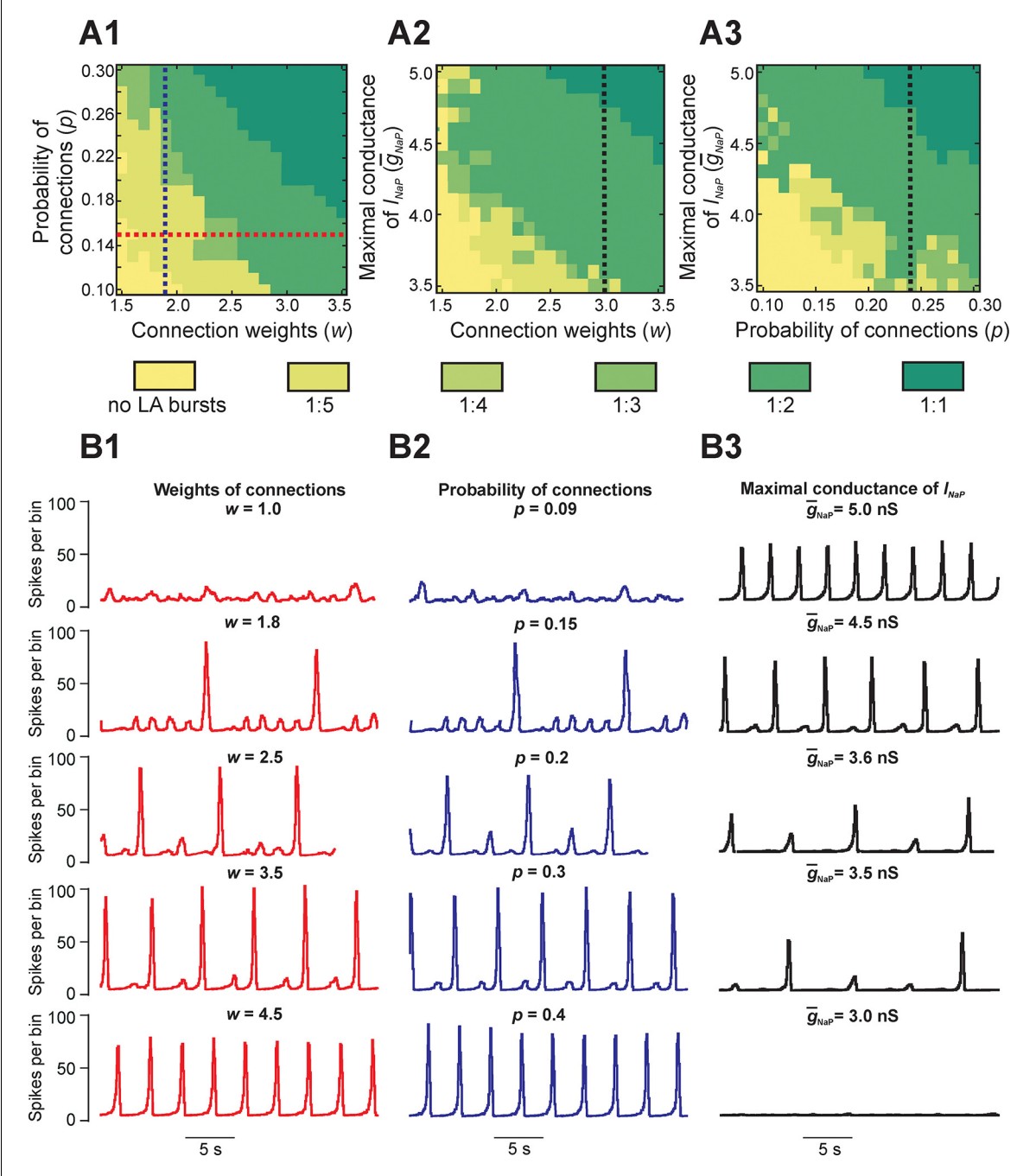

**Figure 4.** Parameter dependence of mixed-mode oscillations. (**A1–A3**) Heat maps depicting quantal changes in the ratio of LA to SA bursts, representing quantal MMO regimes calculated with variation of the connection weights ($w$), probability of connections ($p$), and maximal conductance of the persistent sodium channel ($\overline{g}_{NaP}$). In **A1**, $w$ and $p$ were iteratively varied at $\overline{g}_{NaP}$ = 5 nS. In **A2**, $w$ and $\overline{g}_{NaP}$ were varied at $p$=0.24, and in **A3**, $p$ and $\overline{g}_{NaP}$ were varied at $w$=3. (**B1–B3**) Histograms of population activity (spikes/10ms) were calculated as a parameter of interest was varied. In **B1**, $w$ was varied between 1.0 and 4.5 at $p$=0.15 and $\overline{g}_{NaP}$= 5 nS; these changes correspond to the horizontal red, dashed line in **A1**. Progressive increase of $w$ caused the frequency of LA bursts to increase and the number of SA bursts between LA bursts to decrease. In **B2**, $p$ was varied from 0.09 to 0.4 at $w$=1.8 and $\overline{g}_{NaP}$= 5 nS; these changes correspond to the vertical blue, dashed line in **A1**. Similarly to changes of $w$, increasing $p$ caused an increase in frequency of LA bursts and decrease in the number of SA bursts between LA bursts. In **B3**, $\overline{g}_{NaP}$ was decreased from 5.0 to 3.0 nS, with fixed values $w$=3.0 and $p$=0.24, corresponding to the black, dashed lines in **A2** and **A3**, respectively. This progressive decrease caused a decline in LA burst frequency, and an emergence of SA bursts, until all network activity stopped at $\overline{g}_{NaP}$=3.0 nS.

## Analysis of the quantal nature of MMOs with the reduced model

The reduced model provided an explanation for the emergence of quantal MMOs. A key point was that for each neuron, when it was silent, there was a level of synaptic input that caused its activation. This level depended on the degree of $I_{NaP}$ deinactivation in the neuron, quantified by $h_{NaP}$, as well as on its excitability. When one neuron was activated, it excited the other two neurons, and each of these could be activated if and only if the input it received was sufficiently large (cf. *Rubin and Terman, 2002*). For the *LE* neuron, there were therefore discrete windows of opportunity for activation, corresponding to activation times of the other neurons. This idea can be visualized by considering the trajectory of the full system projected to the ($V_3$, $h_{NaP3}$)-plane (*Figure 5B3,C3,D3,E3*; see *Materials and methods*, *Time-scale decomposition in the reduced model*). When the *LE* neuron is not active, the trajectory evolves along the left branch of the cubic $V_3$-nullcline, corresponding to low $V_3$. The *LE* neuron is activated if the trajectory rises above the left knee, or local maximum, of the $V_3$-nullcline (analogously to the sample trajectory in *Figure 5B3*).

Incoming synaptic excitation lowers the $V_3$-nullcline (*Figure 5C3,D3,E3*), a well-known effect known as fast threshold modulation (*Somers and Kopell, 1993*); the amount of lowering depends on the input strength. In *Figure 5C3*, three $V_3$-nullclines are shown: black corresponds to no input, blue to input from the *HE* neuron only, and green to input from the *HE* and *ME* neurons. If a synaptic input lowers the left knee below the current value of $h_{NaP3}$, then the *LE* neuron is activated (e.g., *Figure 5C3*, marked with 'iv'). Therefore, the activation of the *LE* neuron depends on the recovery of $h_{NaP3}$ when input arrives, and hence on the rate of recovery of $h_{NaP3}$ relative to the frequency of input arrival. For example, in *Figure 5C3*, an SA burst involving only the *HE* neuron occurs when the trajectory is at position 'i'. Since the trajectory is below the knee of the blue nullcline, the *LE* neuron does not activate. An SA burst involving the *HE* and *ME* neurons occurs when the trajectory is at 'ii'. Again, *LE* neuron activation fails, because the trajectory is below the knee of the green nullcline. A failure similar to the first occurs at 'iii'. Finally, when the *HE* and *ME* neurons activate with the trajectory at 'iv', the green nullcline becomes relevant, the trajectory is above the knee, and the *LE* neuron activates, yielding an LA burst.

When synaptic weights were increased, the correspondingly larger excitatory input moved the $V_3$-nullcline to lower $h_{NaP3}$ values, allowing activation of the *LE* neuron with less recovery time (increase of $h_{NaP3}$) and hence with fewer input cycles. *Figure 5D3* shows one *SA* burst without *LE* neuron activation ('i') and one cycle with *LE* neuron activation ('ii'), while in *Figure 5E3*, the *LE* neuron can activate the first time it receives excitation. In all cases, a discrete number of activations of the *HE* and *ME* neurons is needed before $h_{NaP3}$ recovers to a level from which the *LE* neuron can activate (*Figure 5C3,D3,E3*), which gives rise to the quantal nature of the MMO patterns (*Figure 6A*).

The same idea, that activation of a neuron on a specific cycle depends on whether it rises above the knee corresponding to the input it receives, can be used to pinpoint the events associated with transitions between regimes as shown in *Figure 6A*. Since the $h_{NaP}$ value of a knee for a neuron depends on the input level to that neuron (*Figure 5C3,D3,E3*), a curve of knees for the *i-th* neuron can be drawn in the ($input_i$, $h_{NaPi}$) plane (see also *Materials and methods*). Critical connection weights that separate regimes correspond to tangencies to such curves. For example, the transition from 1:1 to 1:2 regimes as *w* is decreased occurs when the *ME* neuron no longer can activate on every cycle (*Figure 5D1*). At the transitional weights, the trajectory, projected to the ($input_2$, $h_{NaP2}$) plane, exhibits a tangency to the curve of knees for the *ME* neuron (*Figure 6C*). Similarly, the next transition, from 1:2 to 1:4, occurs when the *LE* neuron no longer can activate on every second cycle. Thus, at the weights for this transition, the projection of the trajectory to the ($input_3$, $h_{NaP3}$) plane exhibits a tangency to the curve of knees for the *LE* neuron (*Figure 6D*).

In contrast to changes in connection weights, a change in the excitability of the *LE* neuron alone could alter the $V_3$-nullclines (for all input levels) and hence change the frequency of the LA cycles within each quantal MMO rhythm without any change in the overall oscillation frequency of the 3-neuron population (data not shown). On the other hand, an increase in the excitability of the *HE* neuron alone caused an increase of the SA burst frequency. Since the time between SA cycles became shorter, there was less recovery of the *LE* neuron per cycle, such that more SA cycles occurred between LA cycles and the overall LA frequency remained approximately constant (data not shown).

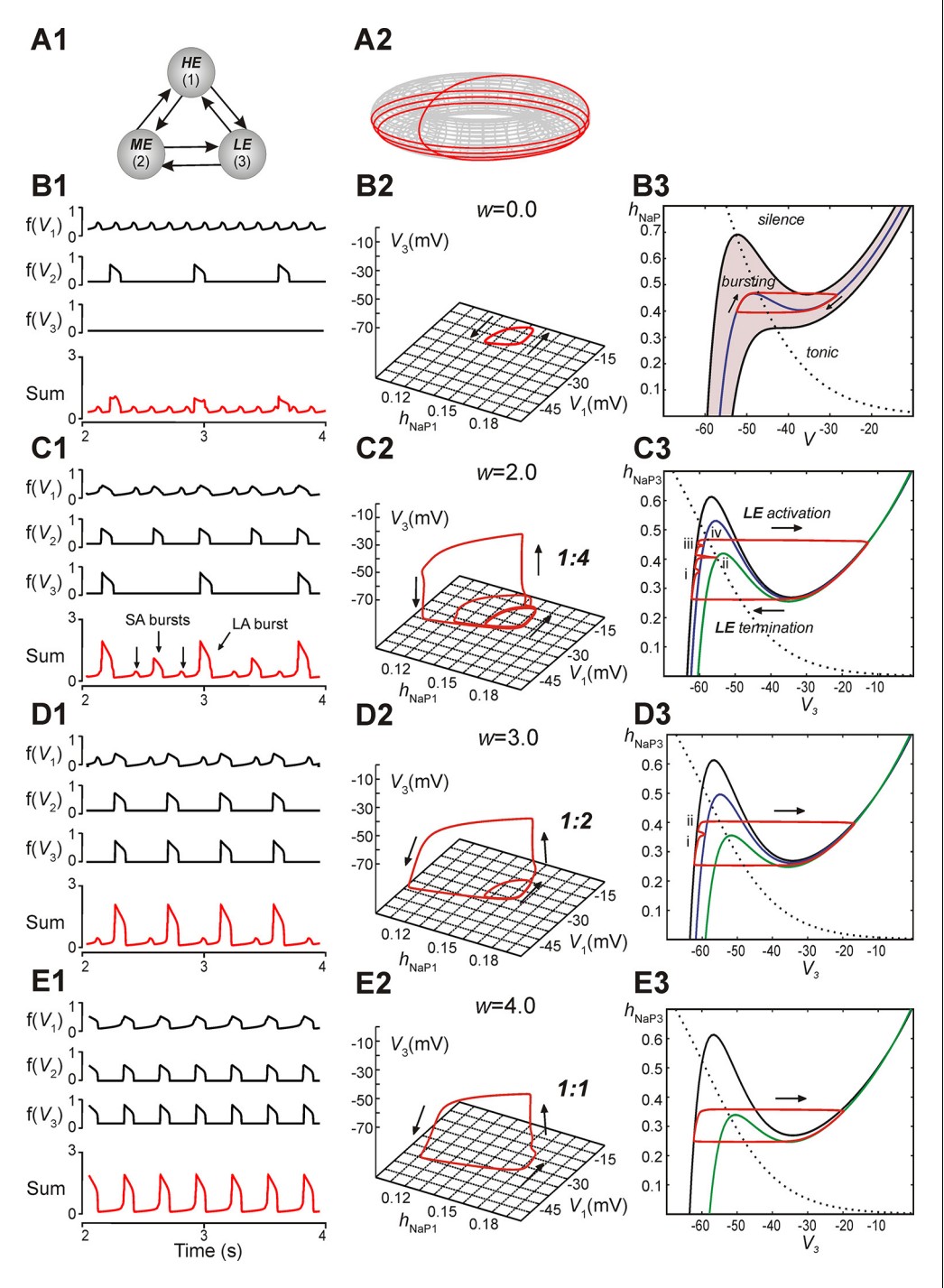

**Figure 5.** Reproduction and analysis of mixed-mode oscillations in a reduced model. (**A1**) Schematic of reduced model with mutual excitatory connections between all neurons. Indices correspond to (1) high-excitability (*HE*), (2) medium-excitability (*ME*), and (3) low-excitability (*LE*) neurons. (**A2**) 1:4 regime represented in toroidal state space (product of two cyclical variables). Four rotations around the larger cycle, corresponding to SA bursts, occur during a single rotation in the smaller cycle, corresponding to an LA burst. Adapted from *Rubin et al., 2011*. (**B1**–**B3**) Simulation results when $w$=0. (**B1**) Output activity, $f(V_i)$, was calculated for each neuron. The 'Sum' trace depicts aggregate network output and is asynchronous when $w$=0 (uncoupled network). (**B2**) A trajectory (red trace) in the $(h_{NaP1}, V_1, V_3)$-plane depicts endogenous *HE* oscillations (cyclical movement in the $(h_{NaP1}, V_1)$-plane), and a silent *LE* neuron (no movement in $V_3$). (**B3**) In the $(h_{NaP}, V)$-plane an endogenously bursting neuron's trajectory (red trace) travels around the local minima and maxima of a *V*-nullcline (blue curve) that intersects the $h_{NaP}$-nullcline (black,

*Figure 5 continued on next page*

*Figure 5 continued*

dotted curve). A band of *V*-nullclines was calculated for the range of $E_L \in$ [-59.0, -53.8] mV where endogenous bursting occurred (gray band). $E_L$ values above and below this range caused tonic activity and silence, respectively. (**C1–C3**) Simulation results when *w*=2. (**C1**) Output activity showed a pattern of three SA bursts between two LA bursts (1:4 quantal regime). LA bursts occurred when all three neurons were active, low amplitude SA bursts occurred when only the *HE* neuron was active, and higher amplitude SA bursts occurred when both *HE* and *ME* neurons were synchronously active. (**C2**) The system's trajectory (red curve) projected into ($h_{NaP1}, V_1, V_3$). Four rotations in ($h_{NaP1}, V_1$) occurred along with only a single rotation in ($V_1, V_3$), denoting an LA burst. (**C3**) The *LE* neuron's trajectory (red curve) is projected into the ($V_3, h_{NaP3}$)-plane. The $h_{NaP3}$-nullcline (black, dotted curve) intersects three $V_3$-nullclines: the black nullcline curve corresponds to *LE* neuron's resting state (no excitatory input), and the blue and green nullcline curves correspond to excitatory inputs from the *HE* neuron and both *HE* and *ME* neurons, respectively. The *LE* neuron receives four inputs, marked (i)-(iv), while at rest. Only input (iv) results in a successful *LE* activation, and therefore an LA burst. (**D1–D3**) Simulation results when *w*=3. (**D1**) Two SA bursts occurred between pairs of LA bursts (1:2 quantal regime). (**D2**) In ($h_{NaP1}, V_1, V_3$) the trajectory makes two rotations in ($h_{NaP1}, V_1$) during one rotation in ($V_1, V_3$). (**D3**) In ($V_3, h_{NaP3}$), the LE neuron receives two excitatory inputs, at points marked (i) and (ii). Nullcline colors are consistent with **B3**. (**E1–E3**) Simulation results when *w*=4. (**E1**) Only LA bursts were observed (1:1 quantal regime). (**E2**) In ($h_{NaP1}, V_1, V_3$), one rotation occurs in ($h_{NaP1}, V_1$) for each rotation in ($V_1, V_3$). (**E3**) The *LE* trajectory is projected into ($V_3, h_{Nap3}$) for the 1:1 regime. The *LE* neuron activates when it receives an excitatory input from the other neurons.

## Reduced model analysis of interburst intervals (IBIs)

Another feature of the MMOs observed in our large-scale model is that IBIs were longer after LA bursts than after SA bursts. This property was seen in the reduced model as well (*Figure 5C1,D1*, and see the multiple values of the period for the *HE* neuron within each quantal regime in *Figure 6A*). The reduced framework elucidates the mechanism underlying this feature. When some neurons are activated, the active neurons excite each other. Each active neuron's variables evolve along the right branch of its *V*-nullcline, and activation ends when they reach the right knee, or local minimum, of this nullcline (see *Figure 5B3*, red trace). Stronger excitation pushes a neuron's *V*-nullcline, including its right knee, to lower $h_{NaP}$ values and hence causes the active phase to end with more $I_{NaP}$ inactivation (i.e., lower $h_{NaP}$-coordinate). Thus, a longer recovery period is needed before subsequent activation of the leading neuron. On LA cycles, all neurons excite each other, which causes a maximal lowering of *V*-nullclines and subsequently yields the longest IBIs.

The difference in post-burst recovery times is evident in the *HE* neuron's trajectory when the 1:2 regime is simulated (*w*=3.0, see *Figure 6B*). The different-size loops shown in ($V_1, h_{NaP1}$) correspond to SA and LA bursts, respectively, and therefore have different maximal $V_1$ and minimal $h_{NaP1}$ values, defined by positions of the $V_1$-nullcline during HE activation. The SA bursts occur due to the *HE* neuron's intrinsic rhythmicity. When the *ME* and *LE* neurons excite the *HE*, the $V_1$-nullcline moves to lower $h_{NaP1}$ and $V_1$ values (lowest green nullcline, *Figure 6B*). This movement extends the active phase by pushing the right knee of the $V_1$-nullcline down. As *ME* and *LE* neuron activity adapts, excitation gradually decreases (green band, *Figure 6B*) but nonetheless, when excitation from *ME* and *LE* neurons is removed, the *HE* neuron returns to the left branch of the $V_1$-nullcline at much lower $h_{NaP1}$ values than following an SA burst. Therefore, the time it takes the *HE* neuron to recover, following an LA burst, is longer than the recovery following an SA burst.

## Effects of reduced neuronal excitability and interconnections

To investigate the dependence of MMO regimes on excitability ($E_L$) we proportionally reduced excitability in all neurons. Quiescence could be induced in the *LE* and *ME* neurons after decreasing all excitabilities by 8% (*Figure 7A*). The frequency of the *HE* neuron decreased, and this produced low frequency SA bursts with no LA bursts. A similar regime of only SA bursts could be produced by decreasing weights of neuronal interconnections (*Figure 7B*). In the example shown, both the *HE* and *ME* neurons participated in the SA bursts. No change occurred in the frequency of the *HE* and *ME* neurons (*Figure 7B*).

The phase diagram in *Figure 5B3* can be used to explain the effects of reduction in neuronal excitability and connection weights. Changing excitability moved the *V*-nullclines corresponding to the unexcited, or resting, state of a neuron. For an uncoupled neuron, increasing $E_L$ caused

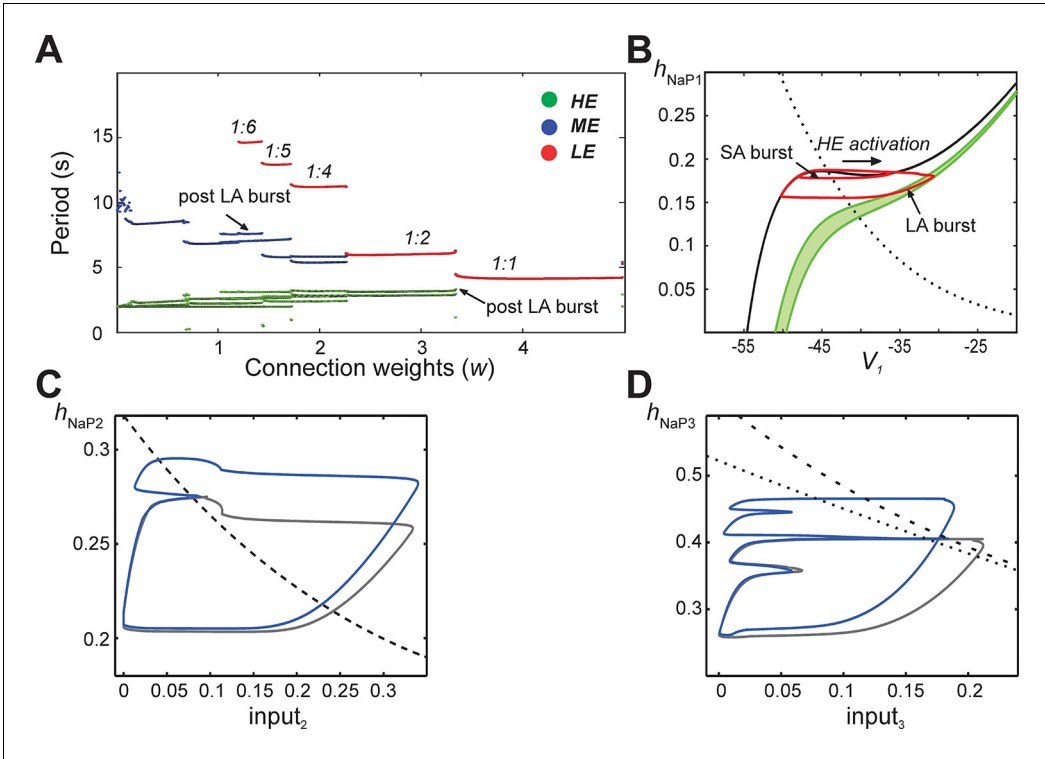

**Figure 6.** Emergence of quantal regimes and analysis of interburst intervals. (A) The burst period of each neuron was continuously calculated as the connection weights ($w$) were increased and neuronal periods on each cycle were plotted. LA bursts occurred at $w>1.4$ (LE emergence, red dots). The quantal regime was determined by the ratio of LE and HE periods. Transitions between stable regimes, i.e. bifurcations, occurred when the LE period 'jumped' to progressively lower integer ratios of the HE period. The ME and HE neurons had longer periods following LA bursts than SA bursts. This phenomenon creates multiple branches in the ME and HE periods for a given quantal regime (see the pair of HE period branches at $w=3$ in the 1:2 quantal regime, for example). (B) The HE neuron's trajectory (red curve) is projected into the ($h_{NaP1}, V_1$)-plane when $w=3.0$ (1:2 regime). Distinct oscillations arise in the HE neuron's trajectory for SA and LA bursts. The black $V_1$-nullcline governs HE activity when it is endogenously bursting during an SA burst. The green $V_1$-nullclines govern HE activity during network-wide activation (LA burst) and are depicted as a band because of the progressive decay of output from LE and ME neurons (resulting from the decrease in $f(V)$ as their voltages decreased, see **Equation 14**, following LA burst onset. (C) Projection of 1:1 trajectory (grey, $w=3.4$) and trajectory at the transition to the 1:2 regime (blue, $w=3.2$) to the ($input_2$, $h_{NaP2}$) plane. The latter exhibits a tangency to the curve of knees (black dashed) of the $V_2$-nullcline, where it fails to activate and thus 1:1 regime is lost. (D) Projection of 1:2 trajectory (grey, $w=2.4$) and trajectory at the transition to the 1:4 regime (blue, $w=2.1$) to the ($input_3$, $h_{NaP3}$) plane. The latter exhibits a tangency to the curve of knees (black dashed) of the $V_3$-nullcline, where it fails to activate and thus 1:2 regime is lost. The curve of fixed points, where the $V_3$-nullcline and $h_3$-nullcline intersect, is also shown (black dotted).

progressive transitions from silence, to bursting, to tonic behavior. The transitions between these behaviors occurred when the fixed point (intersection of the neuron's $V$- and $h_{NaP}$-nullclines) moved from the $V$-nullcline's left branch (silence), to its middle branch (bursting), to its right branch (tonic). When excitability was decreased in a coupled network (**Figure 7A**), the fixed points of the ME neuron moved to the left branch of the $V$-nullcline (the LE neuron's fixed point was already on the left branch, corresponding to the quiescence of the LE neuron in the uncoupled case, see **Figure 5B1**). This decreased excitability increased the amplitude of excitation required to induce bursting in these neurons, and thus the low amplitude HE neuron's phasic excitation was insufficient.

When synaptic weights were changed (**Figure 5B3,C3,E3**) only the $V$-nullclines corresponding to the presence of phasic excitation (from other neurons in the network) were altered. Thus, the intrinsic dynamics of each neuron stayed the same under changes in weights, such that the HE and ME neurons both remained able to activate. With decreased synaptic weights, however, we again found

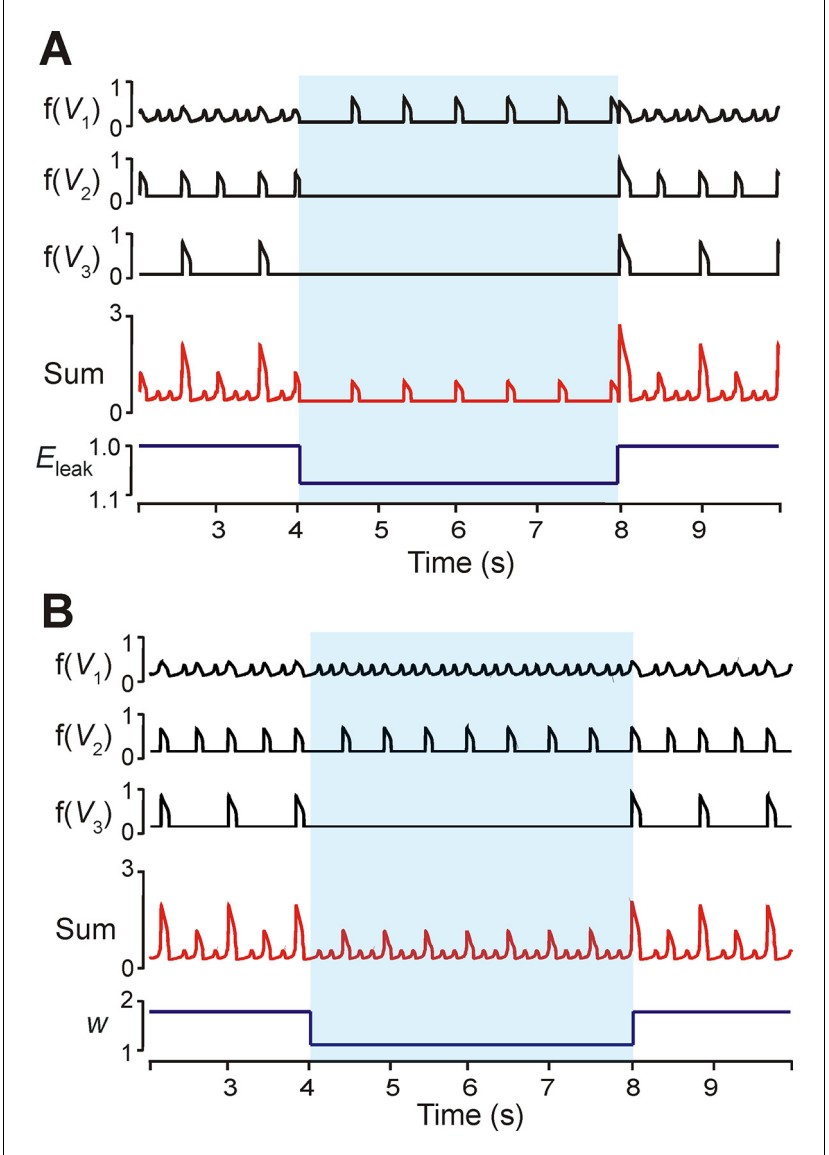

**Figure 7.** Modulation of excitability and connection weights alters reduced model activity pattern. (A) Output levels, $f(V_i)$, for all three neurons with $w=1.7$. A transient decrease of 10% in $E_L$ was implemented between 4 and 8 s (blue shaded region), causing the 1:5 quantal regime to transition to a regime with only $HE$ active. (B) Output levels, $f(V_i)$, for all three neurons with $w=2.0$, producing the 1:4 quantal regime. A transient reduction of $w$ by 50% between 4 and 8 s (blue shaded region) caused a loss of LA bursts. Resulting SA bursts featured activation of $HE$ alone (lower amplitude SA bursts) or synchronized $HE$ and $ME$ activity (higher amplitude SA bursts).

that synaptic excitation could no longer recruit the *LE* neuron (cf. ***Analysis of the quantal nature of MMOs with the reduced model***).

## Discussion

### MMOs in heterogeneous populations of coupled excitatory neurons

We have presented and explored a novel, network-based mechanism for the emergence of MMOs, featuring repetitive alternations of SA and LA bursts of activity, in a heterogeneous population of neurons coupled via sparse excitatory synaptic interactions. In this form of MMOs, the time intervals between bursts are on a similar time scale regardless of whether an SA or an LA burst has just

occurred, yielding quantal patterns of SA and LA events, although precise IBI durations actually depend on the amplitude of preceding bursts, and hence IBIs following LA bursts are longer than those following SA bursts (*Figure 6A*). These MMOs appear to be a natural, perhaps inevitable, behavior of heterogeneous neural networks with excitatory coupling that can be expected to emerge widely in the nervous system, in which the rate of recovery of high excitability neurons dictates the period of subsequent events, while the recovery of low excitability neurons determines which subsequent events become LA bursts. LA bursts correspond to synchronous activation of most neurons in the network and occur when the least excitable neurons in the network can be recruited. Furthermore, feedback from these least excitable to the more excitable neurons is essential for synchronizing the network during LA bursts.

## Relation to MMOs in previous theoretical and modeling studies

The substantial mathematical analysis of synchronization and phase relations in small neural networks with polyrhythmic or multiphase activity has been performed previously without an explicit connection to MMOs (e.g., *Shilnikov et al., 2008*; *Rubin and Terman, 2012*; *Schwabedal et al., 2014*). MMOs have been reported in a variety of neural systems (*Winson, 1978*; *Dickson et al., 2000*; *Medvedev et al., 2003*; *Medvedev and Cisternas, 2004*; *Yoshida and Alonso, 2007*; *Iglesias et al., 2011*; *Golomb, 2014*). The computational and mathematical analysis of these patterns has largely focused on mechanisms that emerge from the separation of time scales typically found within neural dynamics, between voltages and fast gating and synaptic kinetics on one hand and slower gating, synaptic, and ionic concentration kinetics on the other. Within the corresponding MMOs, SA oscillations occur during a delayed transition between two different attractors for the fast dynamics and are often relatively high frequency events that emerge after a quiescent period, whereas the actual transitions between attractors yield LA events (*Desroches et al., 2012*).

Our present work deals with a very different form of MMOs where different oscillation amplitudes correspond to the participation of different numbers of neurons from within a network. In these MMOs, even within SA events, there is a complete transition between different attracting states (hyperpolarized and depolarized) for the fast voltage dynamics, although only some variables in the network are involved in this transition. The MMOs that we studied here depend critically on the synaptic interactions leading to the emergence of neuronal clusters with synchronous bursting activity, whereas the other described classes of MMOs mainly arise from intrinsic dynamics even in single neurons. Therefore, we observed a transition through a range of quantal MMO regimes as synaptic parameters were varied (*Figures 4A,B*,*5*,*6*). Furthermore, LA bursts are gained, as parameters are varied, by conversion of particular SA bursts, arising roughly evenly between pairs of LA bursts, into LA bursts (reminiscent of period-doubling), whereas in time-scale-based MMOs, transitions involve the less radical loss or gain of individual SA oscillations occurring just before each LA burst.

The previous analyses closely related to this novel form of MMO were presented in two earlier papers, both motivated by the pre-BötC in the respiratory brainstem. In one study, synchrony could emerge in a group of modeled neurons with heterogeneous excitability, coupled with synaptic excitation (*Rubin and Terman, 2002*). It was noted that, starting in a 1:1 regime, weakening synaptic strengths could cause less excitable neurons to skip some cycles. In the other previous work, the reduced neuron models were used to investigate quantal recruitment of normally-silent late-expiratory neurons under hypercapnia (*Rubin et al., 2011*). However, the model was not a heterogeneous excitatory network but rather consisted of several distinct neuronal populations coupled with a combination of excitation and inhibition, and the quantal effects observed involved only the single expiratory population, without any clustering or other alterations in other neurons' behaviors.

The other previous study closely related to the present work focused on the dynamic cycle-by-cycle variability in the assembly of neurons contributing to population bursts in the pre-BötC (*Carroll and Ramirez, 2013*). The authors extended previous deterministic models (*Butera et al., 1999b*; *Rybak et al., 2003b*; *Rybak et al., 2004*) by incorporating stochastic drive to all neurons and random, sparse neuronal interconnections. This model could qualitatively reproduce the patterns seen in spike rasters from *in vitro* records. The authors demonstrated the importance of sparse connections in these networks and showed that intrinsically bursting neurons within a sparse network topology play a stochastic, dynamic, and flexible role in the assembly of respiratory rhythms on a cycle-by-cycle basis, which is consistent with our present study.

## Generation of MMOs: the role of endogenous bursting properties of neurons

Despite many years of studies, the exact cellular mechanisms (and ionic currents) responsible for rhythmic bursting in the pre-BötC *in vitro* remain poorly understood and represent a subject of ongoing debate in the literature (*Thoby-Brisson and Ramirez, 2001*; *Del Negro et al., 2002b*; *Del Negro et al., 2005*; *Peña et al., 2004*; *Pace et al., 2007*; *Koizumi and Smith, 2008*; *Krey et al., 2010*; *Dunmyre and Rubin, 2010*; *Beltran-Parrazal et al., 2012*; *Ben-Mabrouk et al., 2012*; *Jasinski et al., 2013*; *Kam et al., 2013*; *Feldman and Kam, 2015*; *Rybak et al., 2014*; *Rubin et al., 2009a*). There are many ionic currents that can be present in pre-BötC neurons and can potentially be involved in population activity. These currents include the persistent (slowly inactivating) sodium current, $I_{NaP}$, (*Butera et al., 1999a*; *Butera et al., 1999b*; *Del Negro et al., 2001*; *Del Negro et al., 2002a*; *Rybak et al., 2003a*; *Rybak et al., 2003b*; *Rybak et al., 2004*; *Koizumi and Smith, 2008*), a calcium-activated, non-specific cation current, $I_{CAN}$, and various $Ca^+$ currents (*Thoby-Brisson and Ramirez, 2001*; *Peña et al., 2004*; *Del Negro et al., 2005*; *Pace et al., 2007*), a transient potassium current, $I_A$ (*Hayes et al., 2008*), and $I_h$ (*Picardo et al., 2013*). $I_{NaP}$ and $I_{CAN}$ have been considered to be the main candidates for currents that are critically involved in pre-BötC bursting. $I_{NaP}$ has been found in pre-BötC neurons and the rhythmic bursting activity in the pre-BötC could be abolished by pharmacological blockade of this current (*Del Negro et al., 2002a*; *Rybak et al., 2003a*; *Rybak et al., 2003b*; *Hayes et al., 2008*) but its critical role in the pre-BötC bursting has been debated (*Del Negro et al., 2002b*). In turn, a series of recent studies of $I_{CAN}$-based bursting in the pre-BötC (*Peña et al., 2004*; *Krey et al., 2010*; *Beltran-Parrazal et al., 2012*; *Ben-Mabrouk et al., 2012*) also produced inconsistent results. *Del Negro et al., 2005* suggested that $I_{NaP}$ may be the primary rhythm-generating current up to postnatal day 4 or 5 (P4 or P5), after which $I_{CAN}$ is also expressed and strongly contributes to rhythm generation. However, all currently available data on MMOs, including the early data (*Koshiya and Smith, 1999*; *Johnson et al., 2001*) illustrated here (see *Figure 1*) and the recent data presented by Kam and Feldman (*Kam et al., 2013*; *Feldman and Kam, 2015*), were obtained, respectively, in slices from the neonatal animals of P0-P2, P0-P3, and P0-P5, i.e. within the developmental range in which $I_{NaP}$ is considered to be the primary rhythm-generating current, supporting the inclusion of $I_{NaP}$ in our models.

In the present work, we studied MMOs in a large-scale neuron population consisting of 100 neurons modeled in the Hodgkin-Huxley style, which were coupled through sparse excitatory synaptic connections. All neurons in the model were capable of endogenous generation of rhythmic bursting activity (*Figure 3A2*) within a particular range of excitability (their resting membrane potential, defined by $E_L$; see *Figures 2A1,A2 and 3A1,A2*). Following the previous computational models of pre-BötC neurons (*Butera et al., 1999a*; *Butera et al., 1999b*; *Rybak et al., 2003a*; *Rybak et al., 2003b*; *Rybak et al., 2004*; *Dunmyre and Rubin, 2010*; *Jasinski et al., 2013*), the $I_{NaP}$ inactivation variable, $h_{NaP}$, evolved with a large time constant and its slow dynamics defined a slow neuronal 'recovery', i.e., gradual depolarization in the post-activity phase (red traces in *Figure 3A2*).

The reversal potential of the leak current ($E_L$) was randomly distributed across neurons in the network to provide a range of excitabilities and subsequent behaviors. This combination of distributed neuronal excitability with slow voltage-dependent recovery provided two important characteristics of neurons within the population:

1. With an increase of excitability in intrinsically bursting neurons, the frequency of bursts increased, whereas the spike frequency within the bursts decreased (*Figure 3A3*); such a reciprocal effect of neuronal excitability on the burst vs. spike frequency arose because with higher burst frequencies (reduced IBIs), there was less time for recovery (deinactivation).
2. Neurons with lower excitability required more time for recovery and could not be involved in high-frequency oscillations.

These two key features of the large-scale model were preserved in our reduced model, in which the spike frequency within the burst was explicitly represented by the amplitude of neuronal output. Therefore, this amplitude decreased with the increasing neuronal excitability (from *LE* to *HE* neurons), and the slow recovery of *LE* neurons (defined by the voltage-dependent time constant for $h_{NaP3}$), was greater than the recovery of *HE* neurons, and prevented the *LE* neuron from participation in higher frequency synchronized bursts (*Figure 5*).

A limitation of our study is that we did not consider burst-generating currents other than $I_{NaP}$. However, although these key features in both of our models result directly from $I_{NaP}$ kinetics, they actually are not specific to the $I_{NaP}$-dependent bursting mechanism analyzed herein. Instead, they represent a common feature of most known cellular bursting mechanisms, in which the post-burst recovery time depends on the neuronal activity within the bursts and vice versa. For example, in the case of intrinsic bursting mechanisms based on $Ca^{2+}$-dependent potassium ($I_K(Ca^{2+})$), $Ca^{2+}$-activated nonspecific ($I_{CAN}$), or $Na^+$-dependent potassium ($I_K(Na^+)$) currents, involving intracellular accumulation of $Ca^{2+}$ or $Na^+$ ions, a functionally similar slow recovery is usually connected with operation of either the $Ca^{2+}$ or Na+/K+ pumps (*Ekeberg et al., 1991*; *el Manira et al., 1994*; *Wallén et al., 2007*; *Rubin et al., 2009a*; *Ryczko et al., 2010*; *Dunmyre and Rubin, 2010*; *Jasinski et al., 2013*; *Rybak et al., 2014*). Therefore the two key features formulated above, which are critical for generation of network-based MMOs, appear to represent common properties of populations of intrinsically bursting neurons with distributed excitability that extend across many different bursting mechanisms. This conclusion clearly contradicts a recently published opinion (*Feldman and Kam, 2015*) that previous computational models reproducing the MMOs observed in the pre-BötC (e.g., *Butera et al., 1999b*; *Rybak et al., 2004*) are not valid because the neuronal bursting in these models is critically dependent on slow deactivation kinetics of $I_{NaP}$.

To evaluate the potential role of $I_{NaP}$ in the considered MMOs, we used our large-scale model to investigate the transition of the population activity pattern during progressive suppression of $I_{NaP}$ in all neurons (*Figure 4A2,A3,B3*). A regime with only LA bursts was selected as a starting point for this study (top trace). When $I_{NaP}$ conductance ($\overline{g}_{NaP}$) was suppressed, the frequency of LA bursts decreased and an MMO regime emerged (*Figure 4B3*, traces 2 and 3) until eventually only SA bursts remained and then activation completely ceased. We consider this result as a prediction for future experimental study, suggesting that a progressive suppression of $I_{NaP}$ in the pre-BötC *in vitro* by its specific blocker, riluzole, should cause a transitional MMO regime before abolishing rhythmicity completely.

## Generation of MMOs: effects of changing connections and neuronal excitability

When the weights of excitatory connections were progressively increased in our large-scale model, a succession of stable network rhythms, or 'regimes', were observed (*Figure 4A1,B1*). Low weights of connections produced only SA bursts in the network's activity (top trace in *Figure 4B1*), intermediate weights caused MMOs (traces 2–4), and strong weights produced regimes with only LA bursts (bottom trace). Similar regimes emerged when the probability of connections was increased at fixed weights of connections (*Figure 4B2*). In all of these cases, the overall frequency of burst events remained similar; what changed was the frequency with which those bursts were of large amplitude.

Similar transformations in the integrated pattern occurred when weights of interconnections were increased in the reduced model (*Figure 5*). In contrast, reduction of either the general neuronal excitability (*Figure 7A*) or weights of connections (*Figure 7B*) could cause *LE* neurons to remain silent, leading to an integrated pattern with only SA bursts present (*Figure 7A,B*). These simulation results may provide a reasonable explanation for the transformation of MMOs observed during application of cadmium ($Cd^{2+}$) in a medullary slice exhibiting MMOs (*Kam et al., 2013*). In these experiments, $Cd^{2+}$ application abolished LA bursts whereas SA oscillations persisted. We therefore suggest that the effects of $Cd^{2+}$, a blocker of calcium currents, could either attenuate neuronal excitability or reduce excitatory synaptic interconnections within the pre-BötC, as seen in our simulations (*Figure 7A,B*). However, more experimental investigations, particularly regarding frequency changes following $Cd^{2+}$ exposure, are needed to distinguish these possibilities.

## The frequency of output pre-BötC oscillations is defined by properties of neurons with the lowest excitability

The analysis of neuronal 'clustering' of our large-scale model showed that groups of neurons with different excitability participated either in SA and LA bursts or only in LA bursts (see *Figure 3B1,B2, C1,C2,D*). Specifically, neurons with relatively high excitability ($E_L$), and therefore with the high burst frequency (*HE* neurons), participated in some SA and all LA bursts, whereas neurons with the lowest excitability and the lowest burst frequency (*LE* neurons) participated only in LA bursts. Importantly,

since *LE* neurons had the highest spike frequency (*Figure 3B3*) within the bursts, they could provide the strongest excitatory synaptic inputs to other neurons, resulting in the network-wide synchronization underlying the generation of LA bursts. It is also interesting to note that *LE* neurons could fail to activate even when receiving excitatory inputs of sufficient strength (see intersection of dashed lines in *Figure 3*D), if the time from the last LA burst was insufficient for the recovery of *LE* neurons. This suggests that a mechanism intrinsic to the *LE* neurons and connected with their slow recovery is critically involved in the generation of LA bursts, defining their IBIs and the output burst frequency.

Our reduced model exhibited a similar dependence on *LE* neuron recovery, which could be confirmed by analysis using time-scale decomposition in the $(V, h_{NaP})$-plane (*Figure 5C3,D3,E3*). This analysis showed that whether or not an excitatory input could recruit the *LE* neuron and induce an LA burst depended on the relative sizes of two quantities: (a) the $h_{NaP}$-coordinate of the *LE* neuron at the time of input (longer periods of recovery, or inactivity, led to higher $h_{NaP}$-coordinates) and (b) the $h_{NaP}$-coordinate of the left knee of the $V_3$-nullcline corresponding to the excitatory input (stronger inputs induced lower $h_{NaP}$-coordinates). Successful *LE* neuron activation occurred when (a) was greater than (b), as at point (iv) in *Figure 5C3*, and activation failed when (b) was greater than (a), as at (i)-(iii) in *Figure 5C3*. When weights were increased, the $V_3$-nullcline was shifted to lower $h_{NaP}$ values, which allowed the *LE* neuron to activate with less recovery.

Interestingly, based on this analysis and previous work (*Dunmyre and Rubin, 2010*) we can infer that the strong mutual excitation, that occurs during an LA burst, is responsible for the pause in activity of the tonic spiking neurons after an LA burst in the large-scale model (*Figure 3B2*). Both the prolonged IBI and the pause in tonic spiking after LA bursts rely on the synaptic excitation from the full collection of neurons in the network, and thus their presence can be taken as evidence that the least excitable neurons in the network are not recipients of feed-forward inputs but rather participate in the recurrent network structure.

## Burstlets, bursts, and separate sub-networks for rhythm and pattern generation

The emergence of MMOs in the pre-BötC has been recently studied *in vitro* in medullary slices from neonatal mice (*Kam et al., 2013*). These MMOs were artificially evoked at a moderate level of neuronal excitability produced by elevation of $[K^+]_{out}$ to 5–6 mM and were characterized by a series of SA bursts ('burstlets') alternating with single LA bursts that, in contrast to the burstlets, were able to trigger the rhythmic bursts in the hypoglossal motor output and hence defined the frequency of output oscillations. This study established the quantal nature of MMOs emerging in the pre-BötC in these conditions (e.g., Figure 2 of *Kam et al., 2013*).

The emergence of these MMOs in the pre-BötC allowed Feldman and Kam to propose a novel 'burstlet concept' of inspiratory rhythm generation that 'fundamentally breaks with the burst hypothesis' (*Feldman and Kam, 2015*). According to this concept, 'rhythm- and pattern-generating functions common to all CPGs are assumed to be segregated' so that the rhythm and the pattern are generated by 'separable microcircuits' and 'distinct mechanisms' (*Kam et al., 2013*; *Feldman and Kam, 2015*), similar to that in a previous model of the spinal locomotor CPG suggesting the existing separate circuits for rhythm generation and pattern formation (*Rybak et al., 2006a*; *Rybak et al., 2006b*; *McCrea and Rybak, 2008*). In this interpretation, the role of intrinsic bursting mechanisms in neurons generating the LA bursts in the pre-BötC is fully disregarded, and the lack of these bursts on the top of each burstlet (SA bursts) is considered as equivalent to the non-resetting spontaneous deletions (missing bursts) observed during fictive locomotion in the spinal cord.

Our computational study does not support the interpretation of MMOs in the pre-BötC as indicative of separate rhythm- (burstlets) and pattern- (bursts) generating sub-networks. The results of our present modeling study instead suggest that a single, inseparable population of coupled excitatory neurons incorporating endogenous neuronal oscillators with distributed excitability can reproduce, and is sufficient to explain, the coexistence of burstlets and bursts in population rhythmic activity (i.e., the MMOs described in this work). We implemented a sparse network connectivity pattern that reflects experimental data more completely than previous models (*Rybak et al., 2004*; *Jasinski et al., 2013*) and precludes the existence of separable sub-networks. In the models of the locomotor CPG in the spinal cord mentioned above, the pattern formation circuits did not affect the rhythm generator circuits, but just responded 1:1 to the rhythm-generating input, unless accidental perturbations happened, changing the excitability of the pattern formation network and

producing deletions (*Rybak et al., 2006a*; *McCrea and Rybak, 2008*). In contrast, in the interconnected single network considered here, the activity of low-excitable neurons involved in generation of low frequency LA bursts (attributed by *Kam et al.* to the 'pattern generating circuits') synchronize the entire population activity, explicitly defining its output frequency ('rhythm'). Therefore, the intrinsic properties of these low-excitable neurons, specifically the temporal characteristics of their recovery (see *Figures 5D1,D2* and *6B*), but not deletions of unknown origin, define the output frequency of a rhythm generator that interacts with other circuits to shape the CPG activity pattern.

## Materials and methods

### Description of single neuron in the large-scale population model

In the large-scale population model all neurons were modeled in the single-compartment Hodgkin-Huxley style, in accordance with our previous models (*Rybak et al., 2003b*, *Rybak et al., 2004*, *Rybak et al., 2007*; *Smith et al., 2007*; *Jasinski et al., 2013*). For each neuron, the membrane potential, *V*, was described by the following differential equation:

$$C\frac{dV}{dt} = -I_{Na} - I_{NaP} - I_K - I_L - I_{SynE}, \tag{1}$$

where *C* is membrane capacitance. The following ionic currents were included in the model: fast sodium ($I_{Na}$); persistent, slowly inactivating sodium ($I_{NaP}$); delayed-rectifier potassium ($I_K$); leak ($I_L$); and excitatory synaptic ($I_{SynE}$). These currents were described as follows:

$$I_{Na} = \overline{g}_{Na} \cdot m_{Na}^3 \cdot h_{Na} \cdot (V - E_{Na}); \tag{2}$$

$$I_{NaP} = \overline{g}_{NaP} \cdot m_{NaP} \cdot h_{NaP} \cdot (V - E_{Na}); \tag{3}$$

$$I_K = \overline{g}_K \cdot m_K^4 \cdot (V - E_K); \tag{4}$$

$$I_L = \overline{g}_L \cdot (V - E_L); \tag{5}$$

$$I_{SynE} = g_{SynE} \cdot (V - E_{SynE}), \tag{6}$$

where $\overline{g}_x$ terms (with index *x* denoting the particular current) represent maximal conductances; $g_{SynE}$ denotes the conductance of the excitatory synaptic current to the neuron (see below); $E_x$ is the current's reversal potential; and $m_x$ and $h_x$ are dynamic variables describing current *x* activation and inactivation, respectively. Activation and inactivation kinetics obey the following equations:

$$\tau_{mx}(V)\frac{dm_x}{dt} = m_{x\infty}(V) - m_x, \tag{7}$$

$$\tau_{hx}(V)\frac{dh_x}{dt} = h_{x\infty}(V) - h_x, \tag{8}$$

where $m_{x\infty}(V)$ and $h_{x\infty}(V)$ define steady-state voltage-dependent activation and inactivation, respectively, and $\tau_{mx}(V)$ and $\tau_{hx}(V)$ are the corresponding voltage-dependent time constants (see *Table 1*). *Equations 1–8* were used for each neuron in the population, with all variables indexed by a numerical subscript specifying the identity of each neuron.

### Interaction between neurons

We considered only excitatory synaptic connections between neurons. The excitatory synaptic conductance was zero at rest and was increased when each excitatory input occurred, such that

$$g_{SynEi} = \overline{g}_{SynE} \cdot \sum_j w_{ji} \cdot \sum_{t_{kj} < t} \exp[-(t - t_{kj})/\tau_{SynE}], \tag{9}$$

where $w_{ji}$ is the synaptic weight from neuron *j* to neuron *i*, $\overline{g}_{SynE}$ is the maximal synaptic conductance, $\tau_{SynE}$ is the synaptic time constant, $t_{kj}$ is the time of the *k*-th spike from neuron *j*, and each term in the sum is evaluated for $t > t_{kj}$. That is, each new spike from neuron *j* increases the excitatory synaptic conductance of neuron *i* by $\overline{g}_{SynE} \cdot w_{ji}$. The probability of each connection (*p*) was set *a priori*, where in a network of *N* neurons, *pN* represents the mean number of neurons with which an individual

**Table 1.** Steady-state functions for voltage-dependent activation and inactivation of ionic channels and other parameter values of the large-scale model.

| Ionic channels | |
|---|---|
| Fast sodium (Na) | $m_{Na\infty}(V) = 1/(1 + \exp(-(V + 43.8)/6.0));$<br>$\tau_{mNa}(V) = 0.25/\cosh(-(V + 43.8)/14.0)$ ms;<br>$h_{Na\infty}(V) = 1/(1 + \exp((V + 67.5)/10.8));$<br>$\tau_{hNa}(V) = 8.46/\cosh((V + 67.5)/12.8)$ ms;<br>$\overline{g}_{Na} = 170.0$ nS. |
| Persistent sodium (NaP) | $m_{NaP\infty}(V) = 1/(1 + \exp(-(V + 47.1)/3.1));$<br>$\tau_{mNaP}(V) = 1/\cosh\left(-(V + 47.1)/6.2\right)$ ms;<br>$h_{NaP\infty}(V) = 1/(1 + \exp((V + 60.0)/9.0));$<br>$\tau_{hNaP}(V) = 6000/\cosh((V + 60.0)/9.0)$ ms;<br>$\overline{g}_{NaP} = 5.0 \ \pm \ 0.5$ nS. |
| Delayed-rectifier potassium (K) | $m_{K\infty}(V) = \alpha_{K\infty}/(\alpha_{K\infty} + \beta_{K\infty});$<br>$\tau_{mK}(V) = 1/(\alpha_{K\infty} + \beta_{K\infty})$ ms;<br>where<br>$\alpha_{K\infty} = 0.01 \cdot (V + 45.0)/(1 - \exp(-(V + 45.0)/5.0));$<br>$\beta_{K\infty} = 0.17 \cdot \exp(-(V + 49.0)/40.0);$<br>$\overline{g}_K = 180.0$ nS. |
| Leak (L) | $\overline{g}_L = 2.5$ nS. |
| Neuron parameters | |
| Reversal potentials | $E_{Na} = 60.0$ mV, $E_K = -94.0$ mV, $E_{SynE} = -10.0$ mV,<br>$E_L = -62.0 \ \pm \ 0.93$ mV. |
| Membrane capacitance | $C = 36.2$ pF. |
| Synaptic/network parameters | |
| Synaptic connections | $\overline{g}_{SynE} = 0.05$ nS, $\tau_{SynE} = 5.0$ ms, $w_{ij} = w \ \in \ [1.0, \ 5.0], \ p \ \in \ [0.09, \ 0.40];$<br>Spike threshold $= -35.0$ mV. |

neuron would form synapses. To form a network, a random number generator was used to determine whether or not each possible synaptic connection among neurons was actually present.

## Simulations

Neuronal heterogeneity within the population was generated with Gaussian distributions for the leak reversal potential ($E_L$) and the maximal conductance of the persistent sodium current ($\overline{g}_{NaP}$). The means and variances of these parameter distributions, as well as all other parameters used in the large-scale model, are provided in *Table 1*.

Initial conditions for neuronal membrane potentials and variables defining currents' activation and inactivation were randomly distributed within physiologically realistic ranges for each variable. To rule out chaotic behaviors, simulations were repeated with redistributed initial conditions for each parameter set. Finally, results were only considered following an initial simulation period of 20 s to minimize the likelihood of transient dynamics.

Integrated population activity was represented by a histogram showing the number of spikes in all neurons per 10ms bin. Maximal values of these histograms during synchronized population bursts, in spikes/bin, were considered as population burst amplitudes. Bursts with amplitude more than 50 spikes/bin were considered to be LA bursts and bursts with amplitude less than 50 spikes/bin were classified as SA bursts.

All simulations were performed using the simulation package NSM 3.0, developed at Drexel University by SN Markin, IA Rybak, and NA Shevtsova. Differential equations were solved using the exponential Euler integration method with a step size of 0.1 ms.

## Generation of heat maps representing MMO dependence of network parameters

To study the effects of changing the connection weights, probability of connections, and the maximal conductance of $I_{NaP}$ on MMO pattern, we calculated the ratio of LA and SA burst frequencies as these parameters were varied (*Figure 4A1,A2,A3*). 50-second simulations were performed, and the

final 40 s were extracted for processing. LA population bursts were defined by histogram activity above 20 spikes in a 100 ms window, and the remaining bursting events were categorized as SA bursts. The ratio of LA and SA bursts was color coded so that boundaries could be visualized in various parameter spaces.

## Reduced model formalization

Mathematical analysis of the large-scale model was prevented by its high dimensionality (100 neurons, each with several differential equations per neuron). However, a preliminary analysis of the simulation results suggested that a minimal neural network could be used to reproduce the development of MMOs caused by the clustering of neurons with similar excitabilities. We therefore developed a reduced network consisting of three neurons simulated by an 'activity-based,' non-spiking model with different excitability defined by the $E_L$ value for each neuron. In this reduced formalization, a neuron's activity represents the aggregate activity of a distinct cluster in the large-scale model. Similar reduced three-neuron models were previously considered in other contexts (e.g., *Shilnikov et al., 2008*; *Rubin and Terman, 2012*; *Schwabedal et al., 2014*). The simplified neuron models have been also previously used to simulate and analyze the behavior of larger models of respiratory networks, including the pre-BötC (*Rubin et al., 2009b*; *Rubin et al., 2011*).

Each neuron is described by one 'fast' dynamic variable, $V$, that governs changes in a neuron's membrane potential and obeys the following differential equation:

$$C \cdot \frac{dV_i}{dt} = -I_{NaPi} - I_{Li} - I_{SynEi}, \tag{10}$$

where $i \in \{1, 2, 3\}$ is the index corresponding to the neuron's number shown in *Figure 5A1* and $C$ is membrane capacitance. This reduced model excluded the fast sodium ($I_{Na}$) and potassium ($I_K$) currents included in the large-scale model. However, similar formalizations of the persistent (slowly inactivating) sodium ($I_{NaP}$), leak ($I_L$), and excitatory synaptic ($I_{SynE}$) currents were used:

$$I_{NaPi} = \overline{g}_{NaP} \cdot m_{NaP\infty}(V_i) \cdot h_{NaPi} \cdot (V_i - E_{Na}); \tag{11}$$

$$I_{Li} = \overline{g}_L \cdot (V_i - E_{Li}); \tag{12}$$

$$I_{SynEi} = \sum_{\substack{j=1 \\ j \neq i}}^{3} (w_{ji} \cdot f(V_j)) \cdot \overline{g}_{SynE} \cdot (V_i - E_{SynE}), \tag{13}$$

where for $x \in \{NaP, L, SynE\}$, $\overline{g}_x$ is the maximal conductance and $E_x$ is the channel's reversal potential, respectively. $E_L$ was uniformly distributed across the 3 neurons in the range [-54.5, -63.5] mV to produce one neuron that was intrinsically quiescent and two that were intrinsically oscillating at different frequencies (*Figure 5B1*); we labeled these as low excitability (LE), medium excitability (ME), and high excitability (HE) neurons. The excitatory synaptic current in *Equation 13* includes inputs to neuron $i$ from neurons $j$, each of which is the product of fixed connection weights, $w_{ji} = w$, and a piecewise linear function, $f(V)$:

$$f(V) = \begin{cases} 0, & \text{if } V < V_{\min} \\ (V - V_{\min})/(V_{\max} - V_{\min}), & \text{if } V_{\min} \leq V < V_{\max}, \\ 1, & \text{if } V \geq V_{\max} \end{cases} \tag{14}$$

where $V_{min}$ and $V_{max}$ define the voltages at which threshold and saturation occur, respectively.

An activity level (or normalized firing rate) for each neuron is implicitly associated with the value of its voltage, and the function $f(V)$ represents an output signal corresponding to that activity level.

The activation of the persistent sodium current, $I_{NaP}$, is described by the voltage-dependent steady state gating variable, $m_{NaP\infty}$:

$$m_{NaP\infty}(V) = \left(1 + \exp\{(V - V_{mNaP})/k_{mNaP}\}\right)^{-1}. \tag{15}$$

$I_{NaP}$ activation is considered instantaneous. The 'slow' dynamical variable in the reduced model, $h_{NaP}$, represents inactivation of the persistent sodium current and is governed by the following equation:

**Table 2.** Parameter values for the reduced model.

| Ionic channels | |
| --- | --- |
| Persistent sodium (*NaP*) | $V_{mNaP} = -40.0$ mV, $k_{mNaP} = -6.0$ mV; $V_{hNaP} = -59.0$ mV, $k_{hNaP} = 10.0$ mV; $V_{\tau NaP} = -59.0$ mV, $k_{\tau hNaP} = 20.0$ mV, $\tau_{hNaPmax} = 5000$ ms; $\overline{g}_{NaP} = 5.0$ nS. |
| Leak (*L*) | $\overline{g}_L = 2.8$ nS. |
| Synaptic Current (*SynE*) | $\overline{g}_{SynE} = 0.1$ nS. |
| **Neuron parameters** | |
| Potentials | $E_{Na} = 50.0$ mV; $E_{L1} = -54.5$ mV, $E_{L2} = -59.0$ mV, $E_{L3} = -63.5$ mV; $E_{SynE} = -10.0$ mV. |
| Membrane capacitance | $C = 20.0$ pF. |
| **Synaptic/network parameters** | |
| Synaptic connections | $w_{ji} = w \in [0.0, 5.0]$. |
| Parameters of output function, *f(V)* | $V_{min} = -50.0$ mV, $V_{max} = 0.0$ mV. |

$$\tau_{hNaP}(V) \cdot \frac{dh_{NaP}}{dt} = h_{NaP\infty}(V) - h_{NaP}, \tag{16}$$

where $h_{NaP\infty}$ and $\tau_{NaP\infty}$ describe the voltage-dependent steady-state and time constant for inactivation, respectively:

$$h_{NaP\infty}(V) = \left(1 + \exp\{(V - V_{hNaP})/k_{hNaP}\}\right)^{-1}; \tag{17}$$

$$\tau_{hNaP\infty}(V) = \tau_{hNaP\text{max}}/\cosh\ \{(V - V_{\tau hNaP})/k_{\tau hNaP}\}. \tag{18}$$

The parameters $V_{xNaP}$ and $k_{xNaP}$ for $x \in \{m,h,\tau h\}$ in **Equations 15,17,18** represent each gating variable's half-activation voltage and slope, respectively.

All parameters of the reduced model were taken from previous works (**Rubin et al., 2009b**; **Rubin et al., 2011**) and are specified in **Table 2**. The distribution of $E_L$ was first set manually to match the large-scale model and then optimized by calculating a series of iterative one-dimensional bifurcation diagrams. The robustness of a given regime (for example, the *LE* period branches marked '1:X' in **Figure 6A**) was determined by the range of connection weights across which the *LE* period maintained an integer ratio to the *HE* period. Simulations were performed and visualized using custom C$^{++}$ scripts and gnuplot, respectively.

## Time-scale decomposition in the reduced model

The complete range of a neuron's dynamics, as a function of $E_L$, was investigated with time-scale decomposition in the $(V,h_{NaP})$-plane (**Figure 5B3**). When projected into the $(V,h_{NaP})$-plane, the dynamical variables, $V$ and $h_{NaP}$, had steady states or 'nullclines' (sets of points for which the right-hand sides of **Equations 10,16**, respectively, were set to zero). Some possible positions of the cubic $V$-nullclines are depicted by a gray band in **Figure 5B3**. The upper and lower boundaries of the band correspond to the lowest and highest values of $E_L$ that produced bursting, respectively. That is, the intersection of the $V$- and $h_{NaP}$-nullclines created a fixed point for the system that, when stable, denotes the point where solutions converge. There were two possible stable fixed points in our model for each neuron: (i) along the left branch of the $V$-nullcline (silence), and (ii) on the right branch of the $V$-nullcline, creating a state of constant depolarization (the activity-based analog to tonic spiking). When $E_L$ was intermediate to values that produced silence and tonic behavior, the $h_{NaP}$-nullcline intersected the $V$-nullcline's middle branch, creating an unstable fixed point with a stable periodic orbit, or oscillation (**Figure 5B3**, red trace), that encompassed the local maximum and minimum of the $V$-nullcline (**Figure 5B3**, blue curve). The presence of a stable periodic orbit corresponded to endogenous bursting in these neurons.

Each periodic orbit has two 'slow' components located close to the neuron's $V$-nullcline and governed by the neuron's $h_{NaP}$ (slow) dynamics, and two 'fast components' connecting between $V$-nullcline branches and governed by the neuron's $V$ (fast) dynamics.. During the slow components, the neuron could be silent or at rest when its trajectory was traveling up the left branch of its $V$-nullcline corresponding to an absence of spike generation, and it could be active or depolarized when its trajectory was traveling down the right branch of its $V$-nullcline, corresponding to spike generation. While at rest, a neuron in the bursting regime slowly 'recovered,' with its trajectory rising to higher $h_{NaP}$-coordinates until it reached the left knee (or fold) of the $V$-nullcline (a bursting neuron, shown in *Figure 5B3*, red trace). At the left knee, a neuron's trajectory moved rightward in the ($V,h_{NaP}$)-plane under the fast dynamics to approach the right branch of the $V$-nullcline, corresponding to activation of the neuron. Once active, the neuron's trajectory traveled downward, to lower $h_{NaP}$–coordinates, along the right branch of the $V$-nullcline until it reached the right knee (fold) of the $V$-nullcline, which caused a leftward jump in the ($V,h_{NaP}$)-plane corresponding to burst termination (*Figure 5B3*, red trace). Similarly, a neuron with a stable fixed point could have slow transient dynamics and be in a rest (active) state as its trajectory traveled along the left (right) branch of its $V$-nullcline.

When a neuron became more excitable, either by an increase in $E_L$ or in its excitatory inputs, the right-hand side of its voltage equation was altered, causing a change in the position of its $V$-nullcline, to a location downward and to the right of the original in the ($V,h_{NaP}$)-plane. Such a change could cause the neuron's fixed point to switch from one branch of its $V$-nullcline to another, yielding a transition from silence to bursting to tonic spiking, depending on fixed point location. This change would also change knee locations; correspondingly, plots can be made showing the $h_{NaP}$-coordinate of the left knee as a function of a parameter or as a function of the input to a neuron in *Equation 13*.

### Link to software

The executable files and scripts used to generate the simulations presented in this manuscript may be downloaded from: http://neurobio.drexelmed.edu/rybakweb/software.htm.

## Acknowledgements

This study was supported by NIH/NINDS grant R01 NS069220 (IAR), NSF grant DMS-1312508 (JER) and in part by the Intramural Research Program of NIH, NINDS (JCS).

## Additional information

### Funding

| Funder | Grant reference number | Author |
|---|---|---|
| National Institute of Neurological Disorders and Stroke | NS069220 | Bartholomew J Bacak Taegyo Kim Jeffrey C Smith Ilya A Rybak |
| National Science Foundation | DMS-1312508 | Jonathan E Rubin |

The funders had no role in study design, data collection and interpretation, or the decision to submit the work for publication.

### Author contributions

BJB, Acquisition of data, Analysis and interpretation of data, Drafting or revising the article, Contributed unpublished essential data or reagents; TK, Acquisition of data, Drafting or revising the article; JCS, Analysis and interpretation of data, Drafting or revising the article; JER, IAR, Conception and design, Analysis and interpretation of data, Drafting or revising the article

### Author ORCIDs

Bartholomew J Bacak, http://orcid.org/0000-0003-0727-3928
Jonathan E Rubin, http://orcid.org/0000-0002-1513-1551
Ilya A Rybak, http://orcid.org/0000-0003-3461-349X

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
