## [Decision Letter]

Thank you for submitting your work entitled "Mixed-mode oscillations and population bursting in the pre-Bötzinger complex" for consideration by *eLife*. Your article has been reviewed by three peer reviewers, one of whom is a member of our Board of Reviewing Editors. The evaluation was overseen by Eve Marder as the Senior Editor.

The reviewers have discussed the reviews with one another and the Reviewing Editor has drafted this decision to help you prepare a revised submission.

Summary:

This study addresses an important issue in rhythm generation: the ability of neuronal networks to operate in a mixed-oscillation mode. Mixed mode oscillations (MMO) are observed as small amplitude population oscillations that alternate with large amplitude population oscillations. The authors clearly demonstrate that these MMOs are an "unavoidable" result of a heterogeneous, and sparsely connected network. The authors studied this phenomenon in the pre-Bötzinger complex (pre-BötC), a relatively small mammalian neuronal network that spontaneously generates respiratory rhythmic activity in isolation.

The authors propose that MMOs are the result of the heterogeneity in excitability properties of neurons, and they use two types of computational models: 1) a 100 cell network based on Hodgkin-Huxley formalisms in which single cells are sparsely interconnected via fast excitatory synapses, and 2) a simplified model of three mutually excitatory, non-spiking neurons. The authors demonstrate that the simplified model exhibits many of the same qualitative/quantitative features of the more complex computational model.

The heterogeneity plays two key roles in the mechanism.

The first key role of heterogeneity is based on the qualitative difference of the endogenous excitability regimes of neurons in the population, which underlies the activity pattern of silent neurons, fast bursting neurons and slow bursting tonic neurons. The mechanism explains the appearance of the large amplitude oscillation by recruitment of the large group of endogenously silent neurons. The variability of the oscillatory excitatory input into the group of silent neurons, i.e. the second key mechanism is explained by the heterogeneity of bursting frequencies in the endogenously bursting neurons. The mechanism was also theoretically explained with a simple model of three reduced neuron-like models representing three subpopulations (silent neurons, fast bursting neurons, slow bursting neurons).

The study is very timely and in part controversial since it offers an alternative explanation to an influential study that suggested that mixed mode oscillations are the result of a distinct subthreshold rhythm generating mechanism (burstlets) which recruits supratheshold bursts. Here the authors state that the alternation between the large and small bursts emerges from the same rhythm generating mechanisms as a result of sparse connectivity combined with the heterogeneity of intrinsic membrane properties. The conclusions presented in this manuscript are of general relevance, since sparse connectivity and heterogeneity of membrane properties are a common property of many neuronal networks that consequently do also generate mixed mode oscillations.

The reviewers raised several comments that should be addressed in a revision.

Essential revisions:

1) The authors are using a reduced conductance set and focusing on persistent sodium as their rhythm-generating current when this method of rhythm generation (in the animal models examined so far) occurs over a very circumscribed developmental range. This has been pointed out in Del Negro et al. (J Neurosci, 2005, PMID:15647488). The persistent sodium current appears to be the primary rhythm-generating conductance from embryonic development and the initiation of breathing rhythm (about embryonic day 15) through postnatal day 4 or 5 in mice and rats. After postnatal day 5, the calcium-activated, non-specific cation conductance (*I_CAN_*) is also expressed and contributes to respiratory rhythm generation, perhaps more importantly than *I_NaP_*. The authors attempt to brush this aside in a paragraph that begins “Although these key features in both our models…” and suggest that changes in the excitability of the neurons that generate MMOs is a generalizable phenomenon – which we agree with – that may not be dependent upon expression levels of *I_NaP_*. However, the kinetics of *I_NaP_* and *I_CAN_* are sufficiently different to make this a more intriguing computational problem to address in the context of MMOs. Toporikova and Butera (J Comp Neurosci, 2011, PMID: 20838868) developed a relatively simplistic model of pre-BötC neurons that incorporates both rhythm-generating conductances five years ago. Dr. Rubin has a model that includes the *I_CAN_* conductances. The authors will likely try to avoid re-designing their model for this manuscript. But, at the very minimum, the lack of an *I_CAN_* should be discussed as a caveat in the Discussion section.

2) All three reviewers noted that this study fails to properly consider prior studies that are of direct relevance for the present manuscript. The Introduction does not provide a broad overview of the current theoretical and experimental state of the field. The same applies to the Discussion: it is very limited to their own work. The authors are probably aware that there is a large amount of computational literature on subnetwork oscillations that is not considered. The following publications need to be considered in this study.

a) There has been considerable work done on mixed-mode oscillations by Shilnikov who refers to the same phenomenon as "polyrhythmic oscillations". Studies include "subthreshold oscillations in a map-based neuronal model" or "Polyrhythmic synchronization in bursting networking motifs" and "Robust design of polyrhythmic neural circuits” by Justus T. C. (2014), just to name three papers. These authors have published a very careful bifurcation analysis of the three cell network as modeled here in the so called "reduced model". The bifurcation analysis presented here, is insufficient. The study does not provide a complete mathematical description that could explain the transitions between the different oscillatory regimes of the network. A bifurcation analysis would significantly strengthen the manuscript. It would be best to provide this, but minimally, this should be discussed as a major caveat of this study.

b) An equally important failure to cite and acknowledge are the studies published by Carroll and Ramirez that first introduced the theory of sparseness in this particular excitatory respiratory network (pre-Bötzinger complex), and used the exact same model to explain rhythm generation. Figure 6 in their first study ("cycle-by-cycle…”) shows the consequences of sparse network size and connectivity on the respiratory oscillations, clearly showing mixed mode oscillations (e.g. Figure 6 inset). And they quantitatively demonstrate the dependency of variability on the connection probability (Figure.6C). Figure 4 of the present study is not too different from the Carroll study. The Carroll study should be included into the Discussion.

c) All three reviewers noted a lack of consideration of the physiological and pathophysiological relevance of the mixed mode oscillations. One suggestion, a recent study by Zanella et al. (J. Neuroscience 2014, 34 (1) 36-50) shows the inductions of MMOs after acute intermittent hypoxia and norepinephrine and discusses the clinical implications. Providing a general physiological and/or pathological context is important to make it more interesting for a general readership- remember this is a manuscript for *eLife*. Indeed, because MMOs more likely represent a pathophysiological form of breathing, it is necessary for the authors to provide the reader with more discussion of eupneic breathing and the disturbances for example in sleep disordered breathing.

3) The authors highlight that they introduce a "novel paradigm for MMos generation" that has been observed experimentally and the authors provide published examples of the preBötC activity in control and in the presence of CNQX. The model implies that the nature of MMOs is quantal. However, it is essential to validate this theory, in particular since the example shown is not very quantal. If it is not quantal, then the authors need to include noise into their model. It would be easy to experimentally address this issue (e.g. with CNQX or riluzole) – Dr. Smith has the required expertise. However, we understand that the authors probably want to keep their study purely computational. In this case the author should at least address this possibility in their Discussion, or explore it further in simulations.

4) The authors should provide a better quantification of the model with n=100 neurons. We recommend plotting 3 different 2D graphs such as heatmaps of <nSA> where (x, y) axis are: 1^st^ graph – connection weights, probability of connection; 2^nd^ – connection weights, conductance of persistent sodium current; 3^rd^ – probability of connection, conductance of persistent sodium current.

5) Why do the authors assume variability of *E_L_* in the same tissue? Is there any experimental evidence for such variability? The authors refer to experimental evidence of variability of the resting membrane potential. The authors should define the resting membrane potential of tonic spiking activity and bursting activity. Note, the variability of the resting membrane potential does not imply the variability of *E_L_*.

To better describe the cellular mechanism, it would be helpful to use heterogeneity of leak conductance, *g_L_* instead of *E_L_. g_L_* contributes to the input resistance and its heterogeneity might affect the effectiveness of the mechanism.

6) Please clarify: are the neurons in Figure 2 and Figure 3 ordered by the values of *E_L_*or by excitability? In order to sort the values by excitability the authors should introduce a clear numerical measure for excitability. If GP was varied along with *E_L_, E_L_* alone could not represent excitability. It would be helpful if the authors provide a plot (*E_L_*, GP) with color-coded properties of decoupled activity of each neuron in the population. Moreover, the authors should provide information on the number of endogenously silent, bursting, and spiking neurons in the population. It would also be important to provide information on the distribution of the burst frequencies in the decoupled population.

7) Is it really important that the connectivity is sparse, or was the sparsity used to better reflect the network organization of the pre-Bötzinger complex? But would the mechanism work if the connectivity is all-to-all and the strength of coupling is sufficiently weak?

8) For the general reader it should be clarified whether the bursts are actually larger in amplitude in the LA bursts than in the SA bursts. The size of the amplitude is referred to the size of the integral output. These terms should be clarified and used appropriately in the manuscript.

9) Is the heterogeneity of the properties of endogenously bursting neurons necessary? Could MMO be observed in a population of neurons of three types (spiking, bursting, and silent) with no heterogeneity within each group?

10) The authors suggest that the bursting behavior of *LE* and *ME* neurons depends primarily upon the relative amount of inactivation (*h_NaP_*) but they have not considered or mentioned the role that *I_A_* may play in synchronizing bursting and LIMITING the probability of generating MMOs during breathing (Hayes et al., J Physiol, 2008, PMID: 18258659). This needs to at least be addressed if the authors do not incorporate *I_A_* in their model. Dr. Rybak has extensive experience with developing models with a large complement of conductances. Hence, like in case of the *I_CAN_*, the authors would be able to alter their model.

11) The authors *must* provide links to their simulation software and make it available to the larger scientific community. The authors *must* publish the C++ code and scripts that they used for the reduced model simulations. This can be done by posting the model code and scripts on any appropriate public data base.

---

## [Author Response]

*1) The authors are using a reduced conductance set and focusing on persistent sodium as their rhythm-generating current when this method of rhythm generation (in the animal models examined so far) occurs over a very circumscribed developmental range. This has been pointed out in Del Negro et al. (J Neurosci, 2005, PMID:15647488). The persistent sodium current appears to be the primary rhythm-generating conductance from embryonic development and the initiation of breathing rhythm (about embryonic day 15) through postnatal day 4 or 5 in mice and rats. After postnatal day 5, the calcium-activated, non-specific cation conductance (I_CAN_) is also expressed and contributes to respiratory rhythm generation, perhaps more importantly than I_NaP_. The authors attempt to brush this aside in a paragraph that begins “Although these key features in both our models…” and suggest that changes in the excitability of the neurons that generate MMOs is a generalizable phenomenon – which we agree with – that may not be dependent upon expression levels of I_NaP_*.*. However, the kinetics of I_NaP_ and I_CAN_ are sufficiently different to make this a more intriguing computational problem to address in the context of MMOs. Toporikova and Butera (J Comp Neurosci, 2011, PMID: 20838868) developed a relatively simplistic model of pre-BötC neurons that incorporates both rhythm-generating conductances five years ago. Dr. Rubin has a model that includes the I_CAN_ conductances. The authors will likely try to avoid re-designing their model for this manuscript. But, at the very minimum, the lack of an I_CAN_ should be discussed as a caveat in the Discussion section.*

The reviewers here raised several important issues that require careful clarification. First, the primary objective of our study was not the modeling of the pre-Bötzinger complex (pre-BötC) within a wide developmental range, but in the modeling and understanding of the phenomenon of MMOs emerging in this type of network. We believe that understanding such MMOs can be important for many different rhythm-generating networks in the brain. Second, although we considered this to be a general mechanism, we wanted to develop a realistic model of a real system that generates such MMOs and to analyze it in the context of the recent suggestion that the generation of such MMOs (alternating burstlets and bursts) in the pre-BötC is based on separate rhythm- and pattern-generating circuits (Kam et al., 2013; Feldman and Kam, 2015). In this connection, it is important to note that all of the data showing these MMOs (or periodic alternation of bursts and burstlets) in the pre-BötC, including the early data from Smith's lab (Koshiya and Smith, 1999; Johnson et al., 2001; see our Figure 1) and the recent data presented by Kam and Feldman (Kam et al., 2013; Feldman and Kam, 2015), were obtained in slices from neonatal animals of P0-P2, P0-P3, and P0-P5, i.e. within the developmental range in which, according to the reviewers' reference to Del Negro et al. (2005), the persistent sodium current (*I_NaP_*) appears to be the primary rhythm-generating current. This implicitly validates the use of this current in the present model, and we clarify this point in the revised Discussion.

At the same time, the cellular mechanism (and ionic currents) responsible for rhythmic bursting in the pre-BötC has not been conclusively determined. For example, Del Negro initially believed in the important role of *I_NaP_* (Del Negro et al., 2001; 2002a), then claimed a critical role of *I_CAN_* (Del Negro et al., 2005), and then suggested an involvement of transient potassium *I_A_* (Hayes et al., 2008) and *I_h_* (Picardo et al., 2013). Moreover, although *I_CAN_* was widely studied and even modeled by some people including us (Rubin et al., 2009; Dunmure et al., 2011, Toporikova and Butera, 2011; Jasinski et al., 2013; Rybak et al., 2014; Toporikova et al., 2015), a series of recent experimental studies of *I_CAN_* produced inconsistent results (Pena et al., 2004; Krey et al., 2010; Beltran-Parrazal et al., 2012; Ben-Mabrouk et al., 2012). Therefore, the involvement of *I_CAN_* in pre-BötC rhythmic activity remains controversial. Nonetheless, to address the reviewers' concerns, we have acknowledged this limitation of our study and have significantly expanded the discussion of this issue in the Discussion section entitled “Generation of MMOs: the role of endogenous bursting properties of neurons.”

Finally, as we described in our original Discussion, and mentioned by the reviewers, we believe that these MMOs (and intermittent bursts and burstlets) are not a specific property of an *I_NaP_*-based mechanism, but rather are a common feature of many bursting mechanisms that are characterized by several simple properties. Specifically, as stated in the Discussion, these properties include a slow recovery process after each burst. This can be a slow voltage-dependent de-inactivation of *I_NaP_* (as in our model), or a slow inactivation of *I_K_* (Ca^2+^) or *I_A_* current, or a slow pumping out of Ca^2+^ or Na^+^ ions accumulated during the burst, as in the case of *I_K_* (Ca^2+^) and *I_CAN_* currents (Rubin et al., 2009; Krey et al., 2010; Jasinski et al., 2013; Rybak et al., 2014). Such a recovery process requires a certain interburst interval for a full recovery. Therefore, if a neuron generates bursts at a relatively high frequency (i.e., has relatively short interburst intervals), then the recovery of its excitability between the bursts is limited. This makes its spiking activity within each burst relatively weak, not allowing these bursts to synchronize many other neurons to produce a large population burst. Thus, only neurons with a low background excitability that generate bursts with relatively low frequency have enough time between bursts for a full recovery, leading to generation strong neuronal bursts (with high spiking frequency) that can synchronize many neurons, hence producing large population bursts. This synchronization mechanism can be based on *I_NaP_* as in our model but is not specific to this current. To better express this point, we have expanded the consideration of this issue in the Discussion, section “Generation of MMOs: the role of endogenous bursting properties of neurons.”

*2) All three reviewers noted that this study fails to properly consider prior studies that are of direct relevance for the present manuscript. The Introduction does not provide a broad overview of the current theoretical and experimental state of the field. The same applies to the Discussion: it is very limited to their own work. The authors are probably aware that there is a large amount of computational literature on subnetwork oscillations that is not considered. The following publications need to be considered in this study. a) There has been considerable work done on mixed-mode oscillations by Shilnikov who refers to the same phenomenon as "polyrhythmic oscillations". Studies include e.g. "subthreshold oscillations in a map-based neuronal model" or "Polyrhythmic synchronization in bursting networking motifs" and "Robust design of polyrhythmic neural circuits by Justus T. C. (2014), just to name three papers. These authors have published a very careful bifurcation analysis of the three cell network as modeled here in the so called "reduced model". The bifurcation analysis presented here, is insufficient. The study does not provide a complete mathematical description that could explain the transitions between the different oscillatory regimes of the network. A bifurcation analysis would significantly strengthen the manuscript. It would be best to provide this, but minimally, this should be discussed as a major caveat of this study.*

Similar to that in our reduced model, Shilnikov’s polyrhythmic oscillations emerge in a network of three neurons. However, these neurons were fully activated exactly once per cycle. The patterns of activity in these 3-neuron networks are very different from the mixed-mode oscillations in our simplified model, where neurons do not alternate, but rather demonstrate synchronized activity of integer ratio periodicity. We agree that the Shilnikov group has done extensive analysis of three-neuron circuits like our reduced model and we have added references to some of their works in the Methods section where the model is introduced (“Reduced model formalization”) and in the Discussion section entitled "Relation to MMOs in previous theoretical and modeling studies."

As for our oscillatory regimes, our Figure 6 does show the sequence of stable solutions that emerge when the connection weight parameter is varied, which is an important part of the information contained in a bifurcation diagram. Furthermore, we have now added new panels Figure 6 and Figure 6 to explain the transitions between 1:1 and 1:2, and between 1:2 and 1:4, regimes in our network. These panels and the associated text, which has been added in the Results section entitled “Analysis of the quantal nature of MMOs with the reduced model”, show that transitions occur when a projection of the trajectory is tangent to a certain key curve, the “curve of knees”, corresponding to bifurcations in the voltage nullcline of a particular neuron. These results provide a description of transitions, expressed in a dynamical systems framework.

*b) An equally important failure to cite and acknowledge are the studies published by Carroll and Ramirez, that first introduced the theory of sparseness in this particular excitatory respiratory network (pre-Bötzinger complex), and used the exact same model to explain rhythm generation. Figure 6 in their first study ("cycle-by-cycle*…*) shows the consequences of sparse network size and connectivity on the respiratory oscillations, clearly showing mixed mode oscillations (e.g. Figure 6 inset). And they quantitatively demonstrate the dependency of variability on the connection probability (Figure 6). The Figure 4 of the present study is not too different from the Carroll study. The Carroll study should be included into the Discussion.*

We thank the reviewers for pointing out on this earlier study. A paragraph about the Carroll and Ramirez (2013) study has been included in the Discussion, where it now constitutes the last paragraph of the section "Relation to MMOs in previous theoretical and modeling studies"

*c) All three reviewers noted a lack of consideration of the physiological and pathophysiological relevance of the mixed mode oscillations. One suggestion, a recent study by Zanella et al. (J. Neuroscience 2014, 34 (1) 36-50) shows the inductions of MMOs after acute intermittent hypoxia and norepinephrine and discusses the clinical implications. Providing a general physiological and/or pathological context is important to make it more interesting for a general readership- remember this is a manuscript for eLife. Indeed, because MMOs more likely represent a pathophysiological form of breathing, it is necessary for the authors to provide the reader with more discussion of eupneic breathing and the disturbances for example in sleep disordered breathing.*

To follow this suggestion, we have mentioned in the Introduction the potential relations of MMOs observed in slice *in vitro* to possible respiratory and other disorders in vivo with references to Zanella et al. (2014). However, the system state and respiration in vivo involves many neuronal and non-neural interactions of pre-BötC with other systems not present in the slice. The breathing pattern during eupnea in vivo is also very different from that recorded in slice from XII root. Therefore, even if similar MMO patterns are observed in vivo in some pathological states, they can be based on many unknown interactions and processes not present in slice. Hence it would probably be too speculative to suggest direct relations of these MMOs to disturbances or diseases in vivo. Of course, our point that the MMO mechanism we discuss is a general feature of rhythmic neuronal networks with certain properties holds irrespective of its connection with specific pathologies, and we hope that our work will inspire readers to consider the presence of MMOs in the rhythmic neuronal circuits of most interest to them.

*3) The authors highlight that they introduce a "novel paradigm for MMos generation" that has been observed experimentally and the authors provide published examples of the pre-BötC activity in control and in the presence of CNQX. The model implies that the nature of MMOs is quantal. However, it is essential to validate this theory, in particular since the example shown is not very quantal. If it is not quantal, then the authors need to include noise into their model. It would be easy to experimentally address this issue (e.g. with CNQX or riluzole)* – *Dr. Smith has the required expertise. However, we understand that the authors probably want to keep their study purely computational. In this case the author should at least address this possibility in their Discussion, or explore it further in simulations.*

We agree with this point, and we thank the reviewers for understanding that we want to keep this study purely computational. In that context, the addition of noise or any other manipulations done to the model cannot establish the quantal nature of real recordings. In the future, we plan to focus more thoroughly on the quantal nature of pre-BötC activity. Fortunately, we can meanwhile refer to Kam et al. (2013), where the authors rigorously studied the quantal nature of pre-BötC MMOs, specifically see Figure 2 in their paper. We have included a reference to this work in the context of the quantal nature of pre-BötC activity in the last section of the Discussion.

*4) The authors should provide a better quantification of the model with n=100 neurons. We recommend plotting 3 different 2D graphs such as heatmaps of <nSA> where (x, y) axis are: 1st graph* – *connection weights, probability of connection; 2^nd^*– *connection weights, conductance of persistent sodium current; 3rd* – *probability of connection, conductance of persistent sodium current.*

We thank the reviewers for this suggestion. These 3 heat-maps have been constructed from a series of simulations and incorporated into Figure 4 (new panels A1, A2, A3); these are discussed in the Results section "Parameter dependence of mixed mode oscillations (MMOs)”.

*5) Why do the authors assume variability of E_L_ in the same tissue? Is there any experimental evidence for such variability? The authors refer to experimental evidence of variability of the resting membrane potential. The authors should define the resting membrane potential of tonic spiking activity and bursting activity. Note, the variability of the resting membrane potential does not imply the variability of E_L_. To better describe the cellular mechanism, it would be helpful to use heterogeneity of leak conductance, g_L_ instead of E_L_. g_L_ contributes to the input resistance and its heterogeneity might affect the effectiveness of the mechanism.*

A more realistic representation of leak currents should actually include separate leak currents for each ion type. In this case, there is no single *I_L_, g_L_* and *E_L_*. Instead, there are separate leak currents and leak conductances for each of K^+^ and Na^+^ with the reversal potentials corresponding to these ions (*E_K_*and *E_Na_*, respectively). Thus, the resting membrane potential (*V_0_*) and neuronal excitability can be controlled by changing one or more leak conductances for particular ions, and there is no such free parameter as *E_L_*.

However, we have followed a huge number of previously published works which used a simplification by considering a single, unified *I_L_, g_L_*, and *E_L_*, so that IL=gL⋅(V−EL). Let's now consider a general Hodgkin-Huxley equation:

C⋅dV/dt=−gL⋅(V−EL)−∑igi⋅(V−Ei),

where the *g_i_* represent all other channel conductances, except *g_L_*. Now, assume for simplicity that all other channels including synaptic ones are closed at rest (*g_i_*=0). Then at rest (dV/dt=0 at *V=V_0_*), we should expect that resting potential *V_0_= E_L_*. Though this is not exactly correct, because not all *g_i_*=0 at rest, it is a very close approximation. And you can see that, in the model, the variability of the resting membrane potential (*V_0_*) is directly defined by the variability of *E_L_* and does not depend on *g_L_*. This argument justifies our use of *E_L_* as a parameter to control model neuron variability and excitability.

The variability of *V_0_*, the pre-BötC population has been well established (for example, see Koizumi et al., 2013, Table 2). Based on the above formalism, the variability of *V_0_* in our model is defined by variability of *E_L_*. The reviewers are correct that in real neurons, excitability does depend on the input resistance or leak conductance(s), and its variability can also affect the variability in the network, even without variability in *V_0_*. We note, however, that it is quite unlikely that the incorporation of additional variability in *g_L_* would add anything to the conclusions of this study.

*6) Please clarify: are the neurons in Figure 2 and Figure 3 ordered by the values of E_L_ or by excitability? In order to sort the values by excitability the authors should introduce a clear numerical measure for excitability. If GP was varied along with E_L_, E_L_ alone could not represent excitability. It would be helpful if the authors provide a plot (E_L_, GP) with color-coded properties of decoupled activity of each neuron in the population. Moreover, the authors should provide information on the number of endogenously silent, bursting, and spiking neurons in the population. It would also be important to provide information on the distribution of the burst frequencies in the decoupled population.*

As stated in the second paragraph of the Results section titled "Computational modeling of a network of pre-BötC neurons with sparse excitatory synaptic interconnections," the neurons were ordered by *E_L_*. Although the term “excitability” has been widely used in the computational modeling literature, it cannot be explicitly measured, especially in active and bursting neurons, since it does not represent a fixed quantity but rather varies in time with dynamical changes of conductances and neuron state. Therefore, in light of its relation to resting potential *V_0_*, we thought it most suitable to order neurons by *E_L_* (with constant *g_L_*). We agree that this choice does not highlight the effect of *g_NaP_* dynamics on the neuronal resting potential. At the same time, the applied variability of this conductance was small (5.0 ± 0.5 nS, see Table 1) and did not produce visible changes in the neural distribution, so that rearranging neurons by "resting" membrane potentials was not different from their distribution by *E_L_*.

Plots considering dynamics of this neuronal model across (*E_L_, g_NaP_*) parameter space have been published in previous papers (see for example Figure 7 in Butera et al., 1999a), and hence we do not wish to repeat these results here. The previously published figures clearly show that the small range of *g_NaP_* values used in our study does not significantly affect the number of silent, bursting, and tonically active neurons in the uncoupled population, as in our Figure 2 and Figure 3. Moreover, for the purpose of this study, it did not matter whether the neuron rearrangement was based on *E_L_* or on the "resting" membrane potentials that could be slightly dependent on *g_NaP_* distribution.

Concerning the number of silent, bursting, and spiking neurons in the uncoupled case, in the first section of the Results, we clearly stated that after rearrangement, neurons with number from 1 to 49 were silent, neurons from 50 to 94 generated bursting, and neurons from 95 to 100 were tonically active. This means that we had 49 silent, 45 bursting, and 6 tonically active neurons. Note that with the small variability of *g_NaP_* present in our study (see above), all neurons were potential bursters, although their actual activity patterns (silence, bursting, or spiking) depended on *E_L_* and (in the coupled case) network interactions. We now mention this point in the first section of the Results. Finally, we note that the distribution of burst frequency in the uncoupled case is shown in Figure 3A3, blue line.

*7) Is it really important that the connectivity is sparse, or was the sparsity used to better reflect the network organization of the pre-Bötzinger complex? But, would the mechanism work if the connectivity is all-to-all and the strength of coupling is sufficiently weak?*

As the reviewers suggested, sparsity was included to reflect the physiologically realistic organization in the pre-BötC. At the same time, the newly added heat map plots in Figure 4 add significant information about the role of sparsity in the generality of the MMOs. Figure 4A1 in particular shows that as probability of connections is increased, an MMO regime can be maintained by decreasing connection weights. Also the MMOs of this type have been previously shown in pre-BötC models that simulated networks of neurons with *I_NaP_*-bursting properties and weak all-to-all excitatory connections, see Butera et al., (1999b, Figure 4) and Rybak et al. (2004, Figure 9), although these MMO patterns have not been specifically investigated in an all-to-all network. Therefore, sparsity is not absolutely necessary to produce MMOs. However, the sparsity can affect the realization of various MMO regimes, and the transitions between them (see Figure 4A1, A3, B2).

*8) For the general reader it should be clarified whether the bursts are actually larger in amplitude in the LA bursts than in the SA bursts? The size of the amplitude is referred to the size of the integral output. These terms should be clarified and used appropriately in the manuscript.*

The details of how histograms representing population activity and how LA and SA amplitudes were defined have been added to the “Simulations” section of the Methods.

*9) Is the heterogeneity of the properties of endogenously bursting neurons necessary? Could MMO be observed in a population of neurons of three types (spiking, bursting, and silent) with no heterogeneity within each group?*

Similar MMO regimes could be produced in a simplified model composed of three types of populations, as the Reviewers have suggested. Moreover, this scenario is quite similar to what we demonstrated in our 3-neuron simplified model, in which each neuron represents one such population. At the same time, there is no physiological evidence to suggest the existence of such 3 distinct populations, whereas one population with distributed parameters is supported by experimental findings in the pre-BötC (Carroll et al., 2013). In addition, heterogeneity is required to increase the number of possible quantal regimes, and is thus necessary to reproduce the relevant experimental data (Feldman and Kam 2014; Kam et al. 2013).

*10) The authors suggest that the bursting behavior of LE and ME neurons depends primarily upon the relative amount of inactivation (h_NaP_) but they have not considered or mentioned the role that I_A_ may play in synchronizing bursting and LIMITING the probability of generating MMOs during breathing (Hayes et al., J Physiol, 2008, PMID: 18258659). This needs to at least be addressed if the authors do not incorporate I_A in their model. Dr. Rybak has extensive experience with developing models with a large complement of conductances. Hence, like in case of the I_CAN_, the authors would be able to alter their model.*

Unfortunately, we did not consider *I_A_* current as well as other currents that are probably present in pre-BötC, such as *I_CAN_* and different Ca^2+^ and Ca^2+-^-dependent potassium and sodium currents; see our response to comment (1). All of these currents can, in some way, affect rhythm-generating mechanisms and MMO dynamics. Incorporating all of these currents and investigating their specific roles were beyond the scope of this particular study and can be considered in the future. Nevertheless, we have now mentioned the potential role of *I_A_* current in the Discussion with reference to the Hayes et al. paper.

*11) The authors* must *provide links to their simulation software and make it available to the larger scientific community. The authors* must *publish the C++ code and scripts that they used for the reduced model simulations. This can be done by posting the model code and scripts on any appropriate public data base.*

Software is now available to download at: http://neurobio.drexelmed.edu/rybakweb/software.htm, and we have provided links in the Methods section “Link to software”. Full descriptions and tutorials are forthcoming.